# Genome editing in plants using CRISPR type I-D nuclease

Keishi Osakabe [1✉], Naoki Wada[1], Tomoko Miyaji[1], Emi Murakami[1], Kazuya Marui[1], Risa Ueta[1],
Ryosuke Hashimoto[1], Chihiro Abe-Hara[1], Bihe Kong[2], Kentaro Yano[2] & Yuriko Osakabe[1✉]

Genome editing in plants has advanced greatly by applying the clustered regularly interspaced short palindromic repeats (CRISPRs)-Cas system, especially CRISPR-Cas9. However, CRISPR type I—the most abundant CRISPR system in bacteria—has not been exploited for plant genome modification. In type I CRISPR-Cas systems, e.g., type I-E, Cas3 nucleases degrade the target DNA in mammals. Here, we present a type I-D (TiD) CRISPR-Cas genome editing system in plants. TiD lacks the Cas3 nuclease domain; instead, Cas10d is the functional nuclease in vivo. TiD was active in targeted mutagenesis of tomato genomic DNA. The mutations generated by TiD differed from those of CRISPR/Cas9; both bi-directional long-range deletions and short indels mutations were detected in tomato cells. Furthermore, TiD can be used to efficiently generate bi-allelic mutant plants in the first generation. These findings indicate that TiD is a unique CRISPR system that can be used for genome engineering in plants.

[1] Graduate School of Technology, Industrial and Social Sciences, Tokushima University, Tokushima 770-8503, Japan. [2] Department of Life Sciences, School of Agriculture, Meiji University, Kanagawa 214-8571, Japan. ✉email: kosakabe@tokushima-u.ac.jp; osakabe.yuriko@tokushima-u.ac.jp

Genome editing has developed greatly in recent years by applying CRISPR-Cas (clustered regularly interspaced short palindromic repeats-CRISPR-associated) systems—RNA-based adaptive immune systems in bacteria and archaea that act as defense systems against foreign genetic elements such as viruses. CRISPR-Cas systems comprise two classes (class 1 and 2) consisting of six types (I–VI) and at least 16 subtypes[1,2]. Among them, Cas9 and Cas12a/Cpf1, which belong to class 2 type II and class 2 type V, respectively, are the CRISPR-Cas systems currently utilized most. The guide RNA (gRNA), which consists of a CRISPR RNA (crRNA) spacer and repeat sequence, and Cas effector proteins form complexes that recognize short sequence elements, called PAMs (protospacer adjacent motifs), in the target DNA[3–5]. Base pairing between the crRNA spacer and the complementary target DNA induces Cas nuclease activation for DNA cleavage[6–11].

A class 2 CRISPR-Cas system that contains a single effector protein (type II; Cas9 and type V; Cas12a/Cpf1) has been utilized in genome editing. In contrast, class 1 CRISPR type I—the most abundant CRISPR system in bacteria—has not been well utilized. Class 1 type I CRISPR systems includes multi Cas subunit complex modules, termed "CRISPR associated complex for antiviral defense" (Cascade) (Cas5, Cas6, Cas7, and Cas8 for type I-E) and a target DNA cleavage module (Cas3 for type I-E)[1,2,11,12]. When Cascade and gRNA bind to the target DNA, Cas3 is recruited to, and activated by, the Cascade complex to cleave the target DNA. Generally, type I CRISPR-Cas systems have longer gRNA sequences when compared with Cas9 and Cas12a/Cpf1, which might allow higher specificity as a genome editing tool than that of Cas9 and Cas12a/Cpf1. Recently, it has been shown that type I-E CRISPR-Cas can induce long-range genome deletion in mammalian cells[13–15]. These findings suggest that use of type I CRISPR-Cas could expand genome editing applications with its unique activity of introducing large chromosomal deletions.

We identified a Class 1 type I subtype CRISPR-Cas genomic locus, named type I-D (TiD). TiD is an uncharacterized CRISPR-Cas system and contains a Cas3 effector protein; however, it lacks a nuclease domain. We found Cas10d to be a unique effector protein that possesses a characteristic nuclease domain of the TiD system that is not found in other CRISPR systems. In addition, one of major Cascade factors for PAM recognition, Cas8 homologous protein[16] is also missing in TiD. Therefore, the mode of interference through PAM recognition to target DNA degradation in the TiD system will differ from that of other type I systems. Here, we explored the new Cas effector proteins in CRISPR/Cas TiD in the context of plant genome editing. We identified Cas10d protein as the functional nuclease, and engineered the TiD system to generate mutations in target genomic DNA in tomato cells. TiD-induced mutations included both small insertions/deletions (indels) and long-range and bi-directional DNA deletions in the target genome. These results indicate that TiD-induced mutation patterns differ from those of other CRISPR systems, such as CRISPR-Cas9 (type II) and CRISPR-Cas3 (type I). Furthermore, TiD-mediated targeted mutations were transmitted to the next generation in the genome-edited tomato lines without off-target effects. These findings suggest that novel genome editing tools can be developed using the CRISPR effector modules of the TiD system.

## Results and discussion
### TiD composed of Cas effector proteins with a Cas10d can be used for genome editing.
The CRISPR/Cas TiD locus consisting of eight Cas genes (Cas1d–Cas7d, Cas10d) followed by an array of repeat-spacer units, was identified from *Microcystis aeruginosa*[1,2].

The typical Cas8 gene—the common effector in CRISPR type I-A, B, C, E, and F[2,16]—is missing from the CRISPR/Cas TiD locus of *M. aeruginosa*, predicting different mechanisms of cascade complex stability and in vivo DNA cleavage activity in TiD compared with other type I sub-types (Fig. 1a). To identify the PAM in the TiD system in *M. aeruginosa*, we performed a depletion assay using the negative selection marker *ccdB*. pCmMa567d10 (Supplementary Fig. 7), containing expression cassettes for Cas5d, Cas6d, Cas7d and Cas10d carrying a mutation in the HD-like domain [dCas10d (H177A)] and gRNAs targeted to the *ccdB* promoter, was introduced into *Escherichia coli* strain BL21-AI followed by a PAM library plasmid, pPAMlib-ccdB (Fig. 1b, Supplementary Fig. 9). *ccdB* negative selection revealed the PAM: 5′-GTH-3′ (H = A or C or T) adjacent to the target sequence (Fig. 1b lower panel, Supplementary Data 1 and 2). When pCmMa567 (Supplementary Fig. 8), which carries expression cassettes for Cas5d, Cas6d and Cas7d, was used for screening instead of pCmMa567d10, GTH PAMs were screened out, but the resulting transformants were unstable, and growth of the *E. coli* cells was very weak. These results suggested that Cas10d requires the correct PAM for full repression, and that Cas10d is a functional counterpart of Cas8 for PAM recognition and stabilization. We did not find any similar amino acid sequences shared between the Cas10d and Cas8 protein families.

To detect the genome editing activity of the TiD and Cas10d nuclease function, we then performed the luciferase single-strand annealing (SSA) recombination system using HEK293T cells (Fig. 1c). This system consists of NanoLuc luciferase containing 300 bp homology arms separated by a stop codon and a target gene fragment. First, the human *AAVS1* gene fragment containing the TiD target site (Supplementary Table S2) was used to evaluate the TiD complex using the 35-bp crRNA spacer sequence. In this assay, HEK293T cells were transfected simultaneously with TiD Cas effectors, gRNA, and NanoLuc interrupted with target gene fragment and firefly luciferase expression vectors, and then the luc reporter assay was carried out 72 hr after transfection. Deletion of either Cas3d or Cas10d abolished TiD genome editing activity in the luc reporter assay (Fig. 1d, Supplementary Data 1), suggesting that both Cas3d or Cas10d have essential roles in genome editing activity. In the original CRISPR locus of *M. aeruginosa* strain PCC9808, both 35-bp and 36-bp spacer sequences are used to target specific genomic DNAs; both spacers function in genome editing in human cells (Fig. 1e, Supplementary Data 1). To evaluate the genome editing activity of TiD for plant genes, we next performed the luc assay for several target rice and tomato gene sequences, *SlIAA9* (important in parthenocarpy)[17] and *NADK2* (*OsNADK2*) (Fig. 1f, Supplementary Fig. 1b, Supplementary Data 1). The results showed that there were several targets with GTT or GTC PAM with higher activity in the luc assay than GTA PAM.

### Targeted mutagenesis by TiD in plants.
Next, we constructed TiD expression vectors with plant-cell-specific-promoters for expression of codon-optimized Cas genes and gRNA were employed to induce site-directed mutagenesis in tomato plant: pTiDP1.2, an all-in-one vector, harboring a single *CaMV35S* promoter driving all 5 ORFs of Cas effectors separated by 2A self-cleaving peptide, and pMGTiD20, in which two expression cassettes under two promoters (*CaMV35S* and *Parsley UBIQUITIN 4-2*) are used to express Cas effector genes (Supplementary Fig. 1a). The separated cassettes in pMGTiD20 were designed to eliminate decreasing C-terminal expression levels in the long single cassette for multiple ORFs in pTiDP1.2. gRNAs targeted a 35-bp sequence in the tomato genes, *SlIAA9* and *RIN* (*SlRIN*; involved in fruit ripening)[18]. For the *SlIAA9* gene, the selected

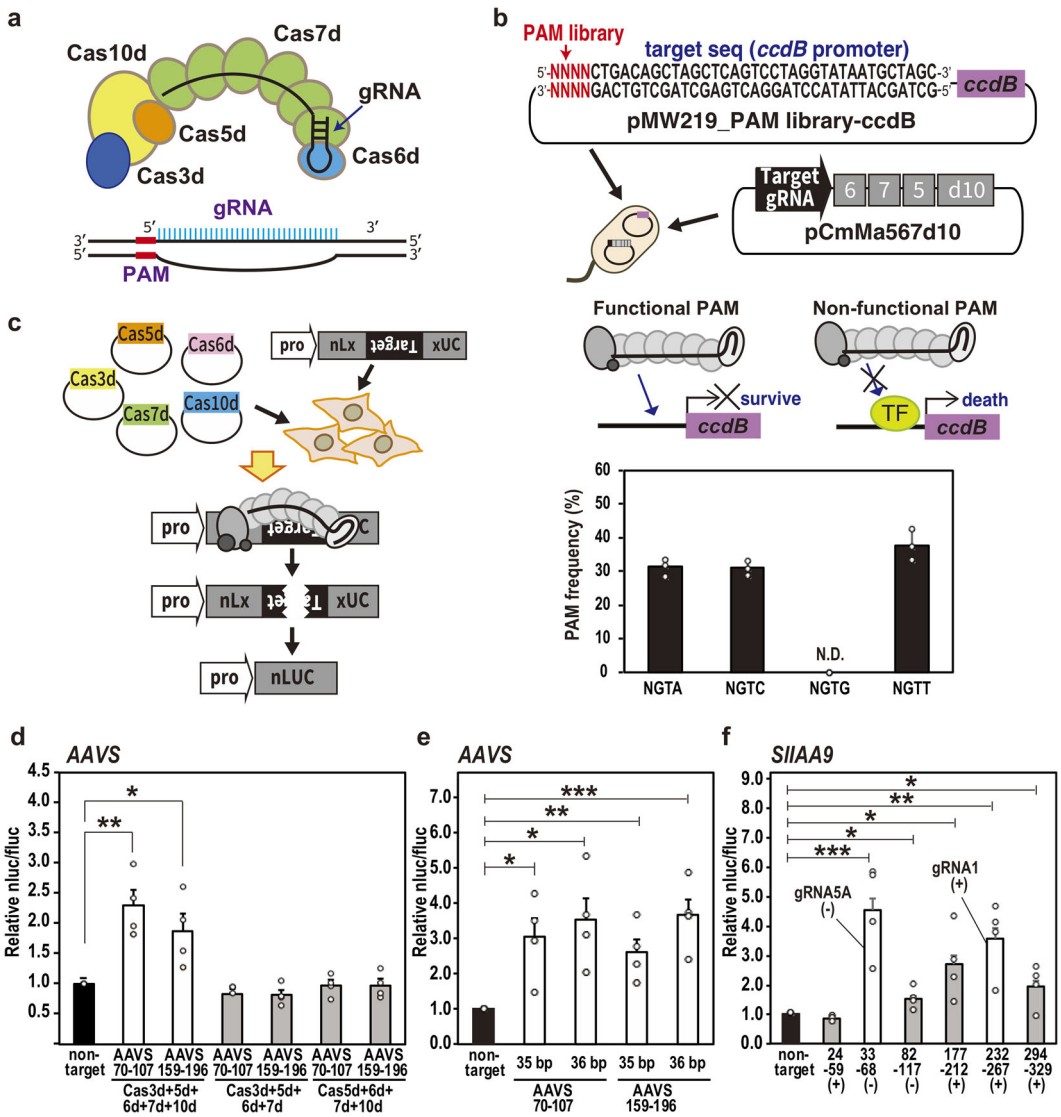

**Fig. 1 Genome editing activity of CRISPR type I-D detected by luc reporter assay. a** The CRISPR type I-D (TiD) structure. Upper; subunit organization of TiD and schematic of gRNA (black) of TiD. Middle; schematic of gRNA (blue) of TiD and target DNA (black). The PAM of the target is shown in red. **b** PAM identification by the *E. coli* negative selection screening using *ccdB* expression system. PAM library was inserted in front of the target sequence of *ccdB* promoter. PAM frequency was determined using the survived *E. colli* cells. Data are means ± S.E. of independent experiments ($n = 3$). **c** Scheme of the luciferase reporter assay used to detect genome editing in human HEK293T cells. The Cas expression vectors and a LUC reporter vector, in which the target sequence was introduced, were transfected into HEK293T cells, and endonuclease cleavage was detected by luminescence. **d** Luc reporter assay showed both Cas3d and Cas10d were required for the TiD activity. Black bar; non-target gRNA in the luc reporter assay. gRNAs were target to the human *AAVS* locus listed in Supplementary Table 1. Data are means ± S.E. of independent experiments ($n = 4$). *$P < 0.05$ and **$P < 0.01$ are determined by Student's t tests. **e**, Effect of the gRNA target sequence length in the TiD activity. Data are means ± S.E. of independent experiments ($n = 4$). *$P < 0.05$, **$P < 0.01$, and ***$P < 0.005$ are determined by Student's t tests. **f** Luc reporter assay to determine targets for the *SlIAA9* gene. gRNAs were target to the tomato IAA9 gene (*SlIAA9*) listed in Supplementary Table 1. Data are means ± S.E. of independent experiments ($n = 4$) and *$P < 0.05$, **$P < 0.01$, and ***$P < 0.005$ are determined by Student's t tests.

gRNAs in the luc reporter assay, GTT_gRNA5-A(−) and GTC_gRNA1(+) (Fig. 1f), both a single gRNA for GTC_gRNA1(+) and multiplex gRNAs for GTT_gRNA5-A(−) and GTT_gRNA5-B(+) (Supplementary Table 1) were used for further analysis. The TiD vector, pTiDP1.2, containing the designed gRNAs, a single gRNA for GTC_gRNA1(+) or multiplex gRNAs for GTT_gRNA5-A(−) and GTT_gRNA5-B(+) were then transformed into the tomato cultivar Micro-Tom by *Agrobacterium*-mediated transformation, respectively. We analyzed TiD-induced mutation efficiency in transgenic tomato calli by Cel-1, PCR-RFLP using AccI[17], and sequencing. In the T0 transgenic tomato calli and shoots, the small indels

mutations were detected by these analyses (Fig. 2, Supplementary Figs. 2, 3).

Cel-1 assay and sequence analysis of PCR products to determine small indels mutations induced by *SlIAA9* GTC_gRNA1(+) revealed somatic mutation in 7/11 transgenic Micro-Tom calli (Fig. 2a, Supplementary Fig. 2a). We further analyzed the mutation efficiency in regenerated tomato shoots using PCR-RFLP for *SlIAA9* GTC_gRNA1(+), identifying undigested bands with AccI in 14/15 transgenic Micro-Tom shoots (Fig. 2b, upper panel, Supplementary Fig. 2b). Together with the sequence analysis, the results indicated that these transgenic shoots contained 100% mutated DNA (Fig. 2e,

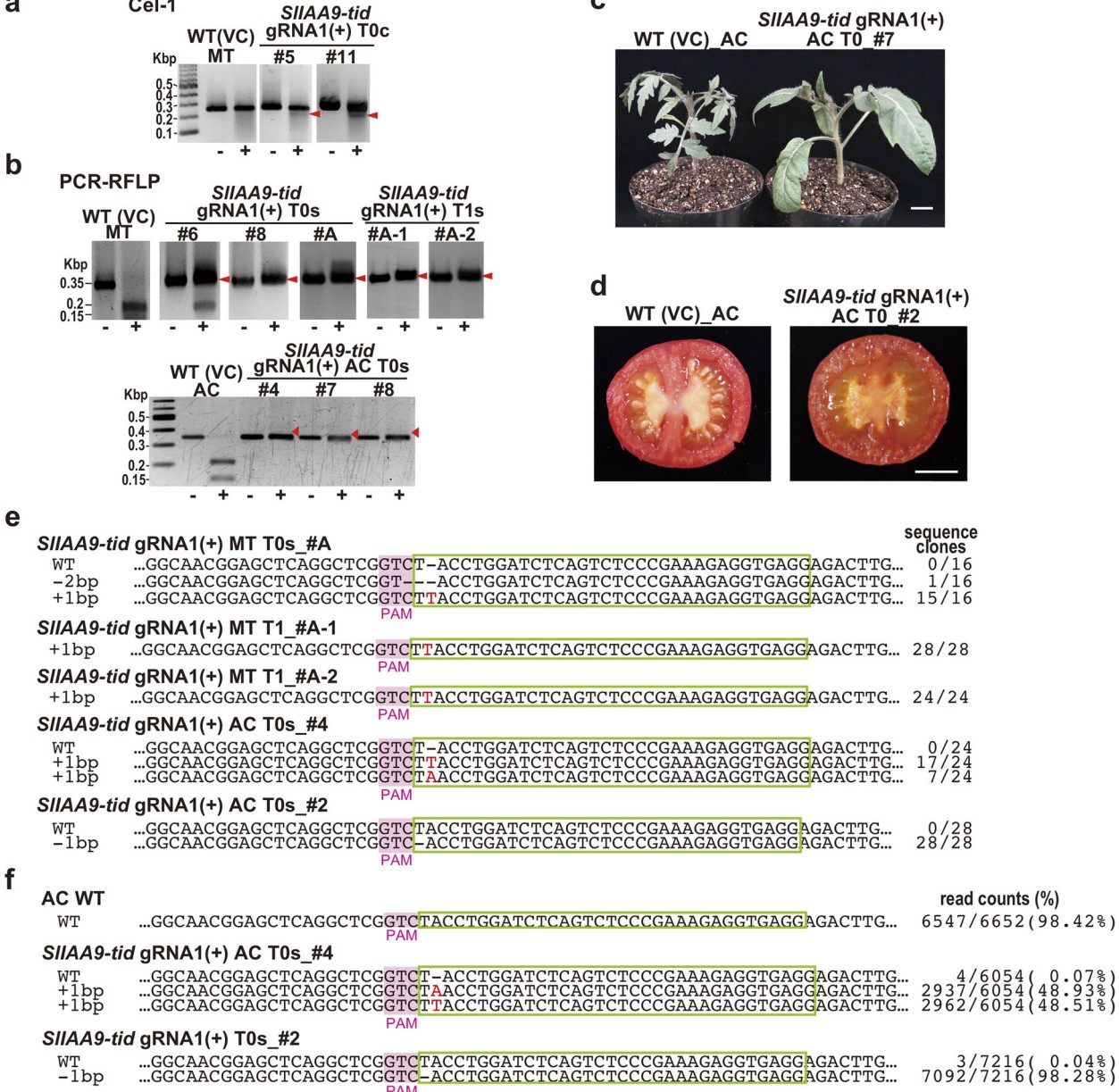

**Fig. 2 Generation of the tomato *IAA9* mutants with small indels mutations by using CRISPR TiD. a, b** The mutations were detected by Cel-1 assay (**a**) and PCR-RFLP (**b**). WT (VC); vector control, MT; Micro-Tom, AC; Ailsa Craig, **a**, #5 and 11; the CRISPR TiD transgenic Micro-Tom calli (T0 generation). **b** Upper, #6, 8, and A; the CRISPR TiD transgenic Micro-Tom shoots (T0 generation), #A-1 and A-2; the next generations (T1) of #A. lower, #4, 7, and 8; the CRISPR TiD transgenic Ailsa Craig shoots (T0 generation), −; without enzymes, +; with Cel-1 nuclease (upper) or AccI (lower), red arrows; mutation bands. **c** Plant phenotypes of *SlIAA9*-disrupted tomato plants (Ailsa Craig) generated by CRISPR TiD. Bars = 2 cm (right). **d** *SlIAA9* knockout tomato fruits (Ailsa Craig) with parthenocarpy phenotypes (right). Bars = 1 cm. **e** Mutation sequences in the *IAA9* gene of Micro-Tom (MT) and Ailsa Craig (AC) mutant shoots (T0 generation) transformed with CRISPR TiD analyzed by the Sanger method. WT; wild-type sequences. gRNA target sequences are indicated in green boxes and PAM is indicated in pink boxes. The sequence frequencies in the cloned PCR products were indicated in the right of the sequence. **f** Mutation sequences in the *IAA9* gene of Ailsa Craig (AC) shoots (T0 generation) transformed with CRISPR TiD analyzed by amplicon deep sequencing using Mi-seq (illumina). WT; wild-type sequences. gRNA target sequences are indicated in green boxes and PAM is indicated in pink boxes. The sequence frequencies in the read counts in the deep sequencing are indicated to the right of the sequence. All results in the electrophoresis and sequence analysis are typical examples from the representative mutant plants generated by TiD.

Supplementary Fig. 3a). Thus, mutation rates were increased in transgenic tomato shoots during regeneration from Micro-Tom calli, and sequence analysis of cloned PCR products from the shoot target DNA revealed bi-allelic mutations. Homozygous mutants were effectively isolated in the T1 generation (Fig. 2b, upper panel, 2e, Supplementary Fig. 3a *SlIAA9-tid* gRNA(+) MT T1_#A-1, #A-2, #B-1). Bi-allelic mutants were also generated

using the commercial tomato cultivar Ailsa Craig, as indicated by PCR-RFLP and sequencing analyses from clone-based sequencing and MiSeq next-generation sequencing (Fig. 2b, lower panel, 2e, 2f). Mature bi-allelic tomato plants exhibited clear typical *SlIAA9* disruption phenotypes, such as parthenocarpy (fertility without seeds) and changes in leaf morphology[17] (Ailsa Craig; Fig. 2c, d, Micro-Tom; Supplementary Fig. 3b).

Together with the *SlIAA9* experiments, TiD induced small indels at the target site (Fig. 2). In the previous study by Ueta et al., the CRISPR/Cas9 that targeted the *SlIAA9* exon2 – located very near the site of the *SlIAA9* GTC_gRNA1(+) – induced biallelic mutations in tomato calli[17]. When comparing the mutation frequencies of CRISPR/Cas9 and TiD in calli, the TiD activities were slightly lower than those of Cas9 (63.6% for TiD and 73.0% for Cas9)[17]; however, the TiD *SlIAA9* GTC_gRNA1 (+) could not induce biallelic mutations in calli (Fig. 1a, Supplementary Fig. 2a). Thus, TiD activity in inducing somatic mutations in calli was lower than that of Cas9. On the contrary, in the shoot samples, TiD could induce biallelic mutations at target sites with efficiency levels similar to those of Cas9 (Fig. 2b, c, Supplementary Figs. 2, 3)[17]. Together, these results suggest there might be tissue specificity in the mechanism of TiD-mediated mutagenesis. Further analyses of other targets will be required to test this hypothesis; for example, detecting mutation patterns in cell lineages during shoot regeneration, and investigating tissue-specific mutagenesis might provide clues to further improvement of the TiD system in plant genome editing.

**Detection of the long-range deletion mutations by TiD in plants.** Type I-E CRISPR-Cas can induce long-range deletion at target sites in the mammalian genome[13–15]. To detect TiD activity in a plant genome, we performed long-range PCR on TiD transgenic tomato calli. Long-range PCR was performed using specific primers located around 2–4 kbp upstream and downstream of the target sequence, respectively (Fig. 3, upper panel, Supplementary Table 4). Figure 3 showed that the several types of long-range deletion induced by *SlIAA9* GTC_gRNA1(+) were detected in transgenic calli by PCR, and sequencing of cloned DNA identified bi-directional deletion (Δ2463 nt) from the mixed PCR product, with a mutation rate of 6.7% (1/15 sequencing clones) in one callus line (#5; Fig. 3, upper-left panel lane 5, Supplementary Fig. 6). Using *SlIAA9* GTT+GTT_gRNA5-(−)(+), specific deletion bands were detected by the nested PCR in 1/20

and 1/30 transgenic calli, respectively (#3; Fig. 3, lower-left panel lane 3, Supplementary Fig. 6). Sequence analysis showed the same 100% mutated fragments in these clones, with bi-directional deletions of Δ4305 nt (Fig. 3). Interestingly, these results indicated that the deletion mutations generated by TiD in tomato genome were bi-directional, which, together with the generation of small indels mutations by TiD, is the unique feature of TiD that differs from mutation by type I-E[13–15], although recent work suggests that Cas9 induced rare complex large deletions in addition to the desired small indels in mouse ES cells, and that these large deletions were bi-directional, similar to TiD but with lower frequency[19]. Furthermore, microhomology and insertions were observed in TiD mutation sites in long-range deletion mutations (Fig. 3), suggesting that specific DNA repair pathways function in these mutations.

Next, we tried to generate a CRISPR TiD targeting another locus in the tomato genome, the *SlRIN* gene, using pMGTiD20 vector, and analyzed mutations in the transgenic calli and regenerated shoots (Fig. 4). Long-range PCR was performed using specific primers located around 3 kbp upstream and downstream of the target sequence, respectively (Fig. 4a, upper panel, Supplementary Table 4). By using *SlRIN* GTC_4003–4238(+), specific deletion bands were detected by the nested PCR in 1/30 transgenic calli (#6; Fig. 4a, lane 6, Supplementary Fig. 6) and the sequence analysis showed the same 100% mutated fragments in these clones, with bi-directional deletions of Δ4930 nt (Fig. 4a). In the analysis for the regenerated shoots, 4 regenerated shoot lines from 12 individual transgenic shoots exhibited specific bands by long-range PCR (Fig. 4b, left panel, Supplementary Fig. 6). Interestingly, similar band patterns were detected in the individual shoot lines #4, #5 and #12 (Fig. 4b, left panel, Supplementary Fig. 6) and sequence analysis of the cloned PCR products indicated two types of long-range deletions (Δ4930 nt and Δ7257 nt); on the other hand, a single type of long-range deletions of Δ7257 was found in line #6 (Fig. 4c, d). The results indicated that the forward PCR primers annealed to homologous sequences 4.6 kbp upstream of the target sequence and could

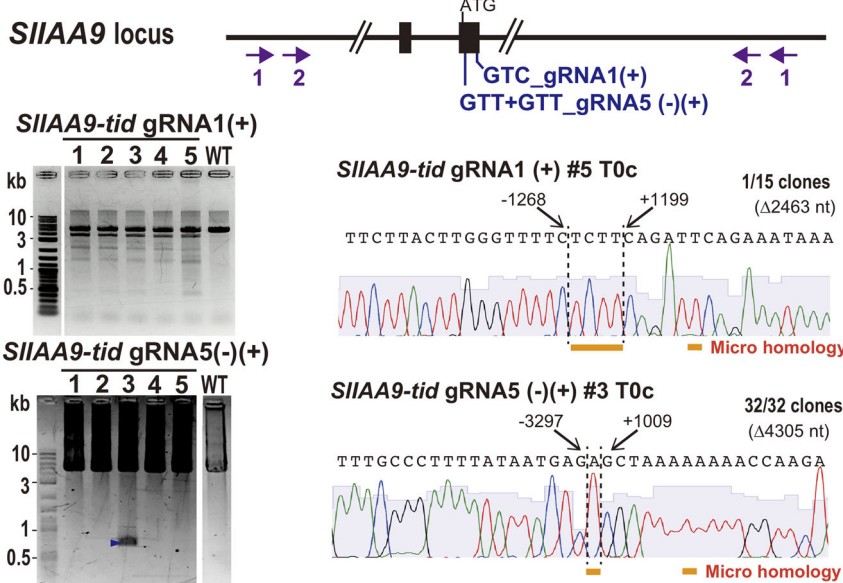

**Fig. 3 Plant genome editing with long-range deletions by using CRISPR TiD.** The detection of long-range deletion mutations in the *SlIAA9* gene induced by the CRISPR TiD. Gene structure, gRNA positions, and the different primer sets to amplify the mutation are indicated. Numbers show the primer sets (1; 1st PCR, 2; nested PCR). The PCR amplified fragments separated on agarose gels are also shown in left panels. The results of large deletion mutations analyzed by the Sanger sequencing of the cloned DNA from the CRISPR TiD transgenic tomato calli are shown in right panels. The nucleotide positions from the PAM were indicated on the sequence. Arrows indicate the specific bands used for the cloning and sequencing. All results in the electrophoresis and sequence analysis are typical examples from the representative mutant plants generated by TiD.

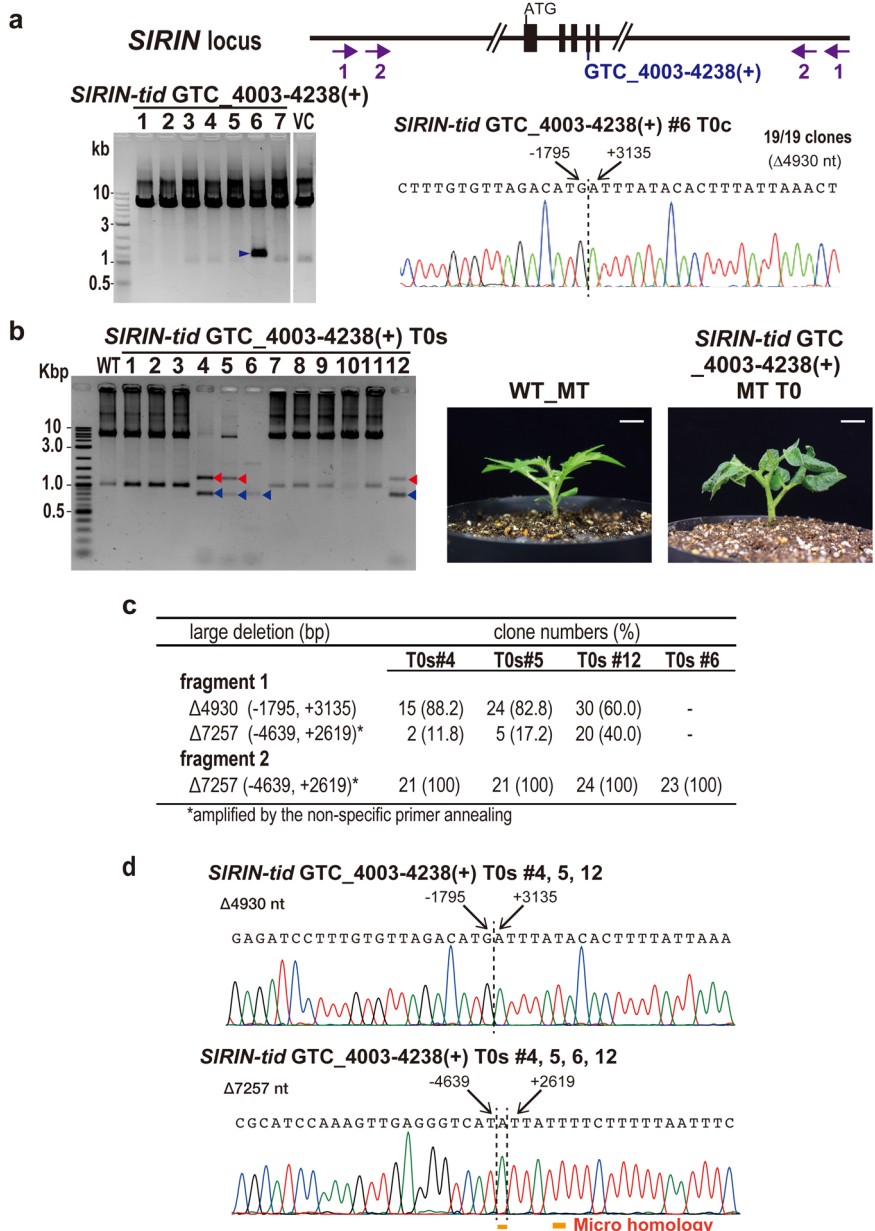

**Fig. 4 Generation of the tomato *RIN* mutants with long-range mutations by using CRISPR TiD. a** The detection of long-range deletion mutations in the *SlRIN* gene induced by the CRISPR TiD. Gene structure, gRNA positions, and the different primer sets to amplify the mutation were indicated. Numbers show the primer sets (1; 1st PCR, 2; nested PCR). The PCR amplified fragments separated on agarose gels indicate CRISPR TiD induced long-range deletions at the tomato *RIN* locus in the mutant calli (Micro-Tom, T0 generation). VC; vector control plants, 1–7; the transgenic callus lines. **b** The PCR amplified fragments separated on agarose gels indicate CRISPR TiD induced long-range deletions at the tomato *RIN* locus in the mutant shoots (Micro-Tom, T0 generation). WT; wild-type, 1–12; the transgenic shoot lines. The large deletions were detected in the lines #4, 5, 6 and 12. The bands with the same length as those of wild-type were non-specific bands. Arrows indicate the specific bands that were subjected to further sequencing analyses; red arrows indicate the fragment1 and blue arrows indicate the fragment2 as shown in **c**. Young mutant shoots for tomato *RIN* (Right; Micro-Tom, T0 generation #6) generated by CRISPR TiD. Bar = 1 cm. **c** Sanger sequencing using the cloned DNA from the CRISPR TiD transgenic tomato shoots (T0; #4, 5, 6, and 12) indicated the large deletion mutations occurred identically, however, the mutation frequencies were varied in the lines. **d** The mutation sequences of the cloned DNA from **a**. The nucleotide positions from the PAM were indicated on the sequence.

detect the Δ7257 mutation in these lines. Together, these results suggest that the bi-allelic mutations were effectively induced by the CRISPR TiD in tomato shoots, and the mature mutant plants for *RIN* were effectively obtained in the first generation (T0) by CRISPR TiD (Fig. 4b, right panel). From these results, we can see that the small indels were not detected in these loci using *SlIAA9* GTT+GTT_gRNA5-(−)(+) and *SlRIN* GTC_4003–4238(+), indicating that varied mutation patterns were induced by each gRNA in the CRISPR TiD system.

**Off-target effects generated by TiD in the plant genome**. We next analyzed TiD off-target effects in plant genome. The TiD targets that has the 5′- GTH -3′ PAM in the whole genome of Arabidopsis and rice, and entire region of tomato chromosome 4 and 5, and each *SlIAA9* and *SlRIN* gene were counted and compared to those of Cas9 (5′- NGG -3′ PAM) (Fig. 5a, b, Supplementary Fig. 4, on-target). In this analysis, tomato chromosomes were selected as being representative of the tomato whole genome. The results indicate the more target sites for TiD

**a**

| mismatch numbers** | SlIAA9 (Solyc04g076850) | | RIN (Solyc05g012020) | |
|---|---|---|---|---|
| | SpCas9 (NGG) | MaTiD (GTH) | SpCas9 (NGG) | MaTiD (GTH) |
| on-target | 421 | 439 | 210 | 461 |
| 0 | 0 | 0 | 1 | 0 |
| 1 | 20 | 0 | 69 | 0 |
| 2 | 263 | 1 | 1149 | 0 |
| 3 | 3544 | 3 | 6512 | 5 |
| 4 | 37568 | 2 | 42129 | 14 |
| 5 | 268883 | 10 | 274934 | 32 |

Header (spanning): Number of target sites*

*number of gRNA target sites for SpCas9 (PAM; -NGG) and MaTiD (PAM; -GTH) were counted.
**mismatch numbers in the gRNA target sequences.

**b**

| mismatch numbers** | tomato ch04 | | tomato ch05 | | Arabidopsis ch01 | | rice ch01 | |
|---|---|---|---|---|---|---|---|---|
| | SpCas9 (NGG) | MaTiD (GTH) | SpCas9 (NGG) | MaTiD (GTH) | SpCas9 (NGG) | MaTiD (GTH) | SpCas9 (NGG) | MaTiD (GTH) |
| on-target | 3694093 | 4836544 | 4576694 | 5318545 | 2024852 | 2518585 | 4516525 | 3376576 |
| 0 | 1567962 | 777154 | 48156778 | 13168095 | 3590875 | 1524655 | 12836488 | 3565284 |
| 1 | 14027408 | 6672036 | 88652431 | 28554442 | 6398057 | 2767420 | 15130824 | 4486074 |
| 2 | 28025517 | 13341425 | 104219138 | 39870254 | 10135564 | 4095706 | 17638958 | 5024562 |
| 3 | 41275187 | 18327809 | 108877203 | 44608558 | 17993819 | 5104391 | 29584075 | 5386008 |
| 4 | 77147142 | 20872970 | 146734054 | 43988678 | 49327434 | 5677644 | 86255454 | 5667482 |
| 5 | 285477726 | 21629094 | 393910236 | 41198459 | 231237832 | 6013954 | 357590050 | 5993240 |

Header (spanning): Number of target sites*

*number of gRNA target sites for SpCas9 (PAM; -NGG) and MaTiD (PAM; -GTH) were counted.
**mismatch numbers in the gRNA target sequences.

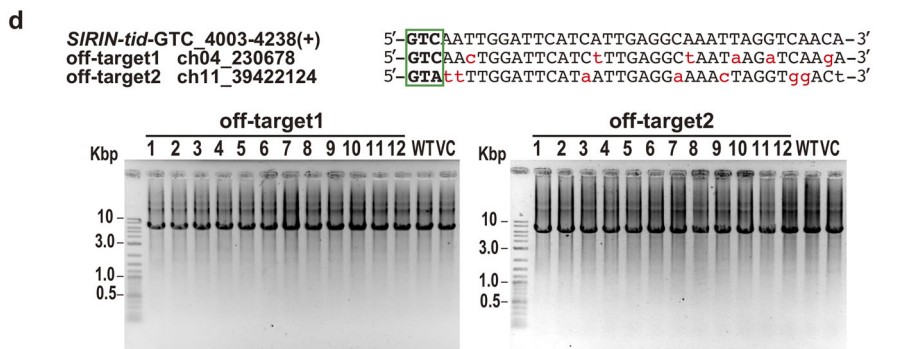

**c**

SlIAA9-tid gRNA1(+)                5′-**GTC**TACCTGGATCTCAGTCTCCCGAAAGAGGTGAGGAG-3′
off-target1 ch09_42659576          5′-**GTC**TACtTGatTaTtAGaCTCCtGAAAGgGGTGAtGAG-3′
off-target2 ch02_35115698          5′-**GTC**TAtaTGGATCTaAGTCggCCGAgcGAtGTcAGGgG-3′
off-target3 ch08_46023555          5′-**GTC**TACaTGGATCTaAGTCggCCaAgtGAtGTtAGGga-3′

| Line No. | mutation frequencies* | | |
|---|---|---|---|
| | off-target1 | off-target2 | off-target3 |
| WT (AC) | 543/43871 (1.24%) | 783/67128 (1.17%) | 1623/51648 (3.14%) |
| SlIAA9-tid gRNA1(+) AC T0s_#1 | 579/55654 (1.04%) | 756/72217 (1.05%) | 1947/53259 (3.66%) |
| SlIAA9-tid gRNA1(+) AC T0s_#2 | 465/47037 (0.99%) | 633/63146 (1.00%) | 1848/34760 (5.32%) |
| SlIAA9-tid gRNA1(+) AC T0s_#4 | 280/34897 (0.80%) | 80/7510 (1.07%) | 2989/38588 (7.75%) |

*mutation efficiencies were caliculated as mutation reads counts / total read counts.

**d**

SlRIN-tid-GTC_4003-4238(+)         5′-**GTC**AATTGGATTCATCATTGAGGCAAATTAGGTCAACA-3′
off-target1 ch04_230678            5′-**GTC**AAcTGGATTCATCtTTGAGGCtAATaAGaTCAAgA-3′
off-target2 ch11_39422124          5′-**GTA**ttTTGGATTCATaATTGAGGaAAAcTAGGTggACt-3′

Fig. 5 Off-target effects of CRISPR TiD in tomato. a, b TiD target site numbers including mismatches in the SlIAA9 and SlRIN genes (a) and the tomato, Arabidopsis, and rice chromosomes (b). The target sites for SpCas9 (PAM; NGG) and MaTiD (PAM; GTH) were counted using Cas-OFFinder and an in-house Perl script. c Upper; The off-target sequences of SlIAA9 GTC_gRNA1(+) target sequence. Red characters; miss-matched nucleotides. Green boxes; PAM. Lower; mutation efficiencies in the cloned PCR products of off-target sites from the transgenic T0 shoots of Ailsa Craig (SlIAA9-tid_gRNA1(+) AC T0s_ #1, 2, and 4) and wild-type (WT) were calculated from the read counts in the deep amplicon sequencing by Mi-seq. d Upper; Off-target sequences of SlRIN GTC_4003–4238(+) target sequence. Red characters; mis-matched nucleotides. Green boxes; PAM. Lower; Long-range PCR (3 kbp + 3kbp) at off-target sites in transgenic T0 shoots of Micro-Tom (SlRIN-tid_ GTC_4003–4238(+)MT T0s_ #1-12), wild-type (WT) and vector control plants (VC).

exist in both the target genes and chromosome levels in tomato and Arabidopsis than those for Cas9. On the contrary, the rice genome has more Cas9 targets than those of TiD, this might result from the higher GC content in the rice genome and Cas9 PAM than in other species and the TiD PAM. Furthermore, when the off-target candidate sequences which contain 0 to 5 mismatches were also counted in tomato whole genome for each SlIAA9 and SlRIN gene, in Arabidopsis and rice whole genomes, and in the representative chromosomes of tomato for the on-targets in the same chromosome, respectively, there are less mismatch sequences for TiD than those for Cas9 (Fig. 5a, b, Supplementary Fig. 4, mismatch numbers 0–5). In rice

chromosomes, the decreasing tendency of TiD targets compared with Cas9 was clearer in off-targets. These data show that there are less off-target sequences for TiD in plant genomes, suggesting a TiD advantage in plant genome editing.

Although the gRNA target sequences used in this study, SlIAA9 GTC_gRNA1(+), SlIAA9 GTT + GTT_gRNA5-(−)(+), and SlRIN GTC_4003–4238(+), do not have highly similar sequences, and have fewer mismatches in the tomato genome, we next evaluated the off-target mutations in the T0 generation of tomato plants exhibiting clear SlIAA9-gene knock-out phenotypes; Three potential off-target sites for SlIAA9 GTC_gRNA1(+) with 9–11 mismatches, two potential off-target sites for SlIAA9 GTT +

GTT_gRNA5-(−)(+) with 11 mismatches, and two potential off-target sites for *SlRIN* GTC_4003–4238(+) with 6 and 7 mismatches, respectively, which are the sites with lowest mismatches for each on-target, were selected and further analyzed (Fig. 5c, d, Supplementary Table 5, Supplementary Figs. 5, 6). MiSeq analysis of PCR products around the potential off-target sites for *SlIAA9* GTC_gRNA1(+) showed that there was little-to-no off-target mutation in the T0 generation of tomato plants (Fig. 5c). Long-range nested PCR of the potential off-targets for *SlIAA9* GTC_gRNA1(+) and *SlIAA9* GTT + GTT_gRNA5-(−)(+) was also performed using specific primers located around 5–8 kbp upstream and downstream of the target sequence, and the results suggested there were no obvious effects (Supplementary Fig. 5a, b). Also, the off-target effects of long-range deletion mutations were evaluated for *SlRIN* GTC_4003–4238(+) in the T0 transgenic plants using specific primers located around 3 kbp upstream and downstream of the target sequence, respectively. As before, no off-target mutations were found in the T0 generation of tomato plants (Fig. 5d, Supplementary Fig. 6). The Cel-1 assay to evaluate small indels also showed no digested bands in the *SlIAA9* GTT + GTT_gRNA5-(−)(+) and *SlRIN* GTC_4003–4238 (+) lines, indicating no mutations in these off-target sites (Supplementary Fig. 5c). Together with a comprehensive analysis of many other on-targets for TiD, further work in vivo to evaluate off-target effects for fewer mismatches will be required in order to precisely elucidate the mechanisms of the TiD system when used in conjuction with the advanced unbiased technologies, i.e. CIRCLE-seq[20] and DISCOVER-seq[21].

**Conclusions.** Although there are eight subtypes of CRISPR type I families identified from bacteria and archaea[2], type I-D CRISPR-Cas, namely TiD, remains less well characterized. We showed that the CRISPR/Cas TiD locus from *M. aeruginosa* strain PCC9808 consists of eight Cas genes (Cas1d–Cas7d, Cas10d) followed by an array of 36 repeat-spacer units. In the TiD system, the HD domain, a functional DNA cleavage domain that has been identified in CRISPR type I-A, B, C, E, and F[7,22–26], is lacking in Cas3d. Instead of the active Cas3 nuclease, TiD has Cas10d, which has an HD-like nuclease domain in the N-terminal region[1]. Interestingly, the Cas10d in TiD was highly divergent compared with Cas10s in the type III CRISPR-Cas family; instead, the Cas10d HD domain was similar to the Cas3 HD domains of type I- B, C, E, and F[1]. In the present study, we first developed a CRISPR TiD system as a genome editing tool for site-directed mutagenesis yielding both short indels and long-range deletion mutations in plant cells. Notably, the desired phenotypes in TiD transgenic mutated tomato were identified. Plant genomes, especially those of crop plants, have complex genome gene structures, with highly duplicated and redundant functional genes, as well as clusters of miRNA and non-coding RNA regions. The specific features of CRISPR type I-D CRISPR, which produces diverse and long-range deletion in genomic regions of interest, could be an effective genome editing tool kit with which to remove complex genome gene structures with low off-target effects. In this study, we used two types of TiD vectors, both of which induced mutations at their respective targets in the tomato genome. Further improvement of the TiD vector will still be important in developing this efficient tool. The unique TiD-induced mutation patterns suggest that the specific DNA cleavage mechanism and subsequent DNA repair pathway may differ from those of other genome editing tools. The diverse range of large deletions that can be generated from a single target site by TiD would enable long-range chromosome engineering; thus allowing expansion of the types of plant genome engineering that are possible using novel technologies in the CRISPR-Cas system.

## Methods

**Vector construction.** All plasmid DNAs used in this study are shown in Supplementary Figs. 7 to 22. In the construction of plasmid DNAs, PCR amplification for cloning was carried out using PrimeSTAR Max (TaKaRa), cloning for assembling was performed using Quick ligation kit (NEB), NEBuilder HiFi DNA Assembly (NEB), and Multisite gateway Pro (Thermo Fisher Scientific).

*Bacterial vectors.* Gene fragments corresponding to *E. coli* codon-optimized *Cas* effector genes consisted of Cascade; *Cas5d*, *Cas6d*, *Cas7d* and *dCas10d (H177A)* the expression cassette for gRNA consists of the crRNA spacer corresponding to the target *ccdB* promoter sequence flanked on both sides by the 37-bp CRISPR repeat, were synthesized (gBlocks®) (IDT), assembled, and cloned into the pACYC184 vector (Nippon Gene) separately. Expression of *Cas* genes and gRNAs was driven by the T7 promoter. For gRNA expression, a DNA fragment containing a T7 promoter-repeat-spacer-repeat sequence was cloned into pACYC184. After confirming sequences of each gene expression cassette, *Cas* gene and gRNA cassettes were re-assembled into pACYC184 to yield pCmMa567 containing *Cas5d*, *Cas6d*, *Cas7d*, and gRNA expression cassettes, and to yield pCmMa567d10 containing *Cas5d*, *Cas6d*, *Cas7d*, *dCas10d (H177A)*, and gRNA expression cassettes as shown in Supplementary Fig. 7. For PAM screening, the PAM screening reporter plasmid was constructed, assembling the sequence of the *lacI* gene, the *lacI* promoter to the 129th codon of the *lacZ* gene following the *ccdB* gene in pMW219 (Nippongene) to yield pPAM-ccdB. To generate a PAM screening reporter plasmid library pPAMlib-ccdB, the gene fragment corresponding to the T7 promoter, lac operator, and PAM with 4-nt randomized nucleotide was synthesized (IDT) and inserted in front of the *lacZ-ccdB* gene of pPAM-ccdB.

*Mammalian vectors.* Plasmid DNAs for the luc reporter assay were constructed using gene fragments corresponding to human codon-optimized *Cas* effector genes; *Cas3d*, *Cas5d*, *Cas6d*, *Cas7d*, and *Cas10d*, were synthesized with the SV40 nuclear localization signal (NLS) at their N-termini (gBlocks®) (IDT), assembled, and cloned into the pEFs vectors[27].

*Luc reporter assay plasmids.* The NanoLUxxUC expression vector was constructed for the luc reporter assay. First, DNA fragments of "NLUxxUC_Block1" and "NLUxxUC_Block2" were synthesized (IDT). "NLUxxUC_Block1" includes the 5′ end of the *NanoLUC* gene (351 bp) and Multi Cloning Site, and an XbaI site was attached to the 5′ end of the *NanoLUC* gene. "NLUxxUC_Block2" includes 465 bp of the 3′ end of the *NanoLUC* gene and an XhoI site was attached to the 3′ end. "NLUxxUC_Block1" and "NLUxxUC_Block2" fragments were then assembled and replaced with EGxxFP fragments in the pCAG-EGxxFP vector (Addgene; #50716).

*Plant vectors.* Gene fragments corresponding to tomato codon optimized *Cas* effector genes; *Cas3d*, *Cas5d*, *Cas6d*, *Cas7d*, and *Cas10d*, were synthesized with a 2x SV40 nuclear localization signal (NLS) at N-termini (gBlocks®) (IDT), assembled and cloned into the pNEB193 (NEB) vector. After confirming their sequences, Five *Cas* genes were re-assembled by connecting via the 2 A self-cleaving peptide sequence, and the *Cas* genes expression cassette were cloned between the *2xCaMV35S* promoter and the *Arabidopsis HSP18.2* gene terminator of pEgP237-2A-GFP[17] to yield pEgP1.2-TiD. To construct the gRNA expression vector, a DNA fragment containing the repeat-spacer-repeat sequence was cloned under the *Arabidopsis U6-26 snRNA* promoter in pDONR P3-P2 to yield pE(L3-L2) AtU6gRNA. The AtU6 promoter- repeat-spacer-repeat fragment in pE(L3-L2) AtU6gRNA was re-cloned into pEgP1.2-TiD to yield pTiDP1.2. To construct pMGTiD20, intermediate vectors for multisite gateway assembling, pE(L1-L4)P1.2-Cas3d-Cas6d-GFP and pE(R4-R3)Ppubi4-Cas10d-Cas5d-Cas7d were constructed. *Cas3d*, *Cas6d* and *GFP* gene fragments driven by the *2xCaMV35S* promoter, and the *Arabidopsis HSP18.2* gene terminator was cloned into pDONR P1-P4 to yield pE(L1-L4)P1.2-Cas3d-Cas6d-GFP. *Cas10d*, *Cas5d* and *Cas7d* gene fragments driven by the *Petroselinum crispum UBIQUITIN 4-2* promoter (Ppubi4) from pEgPubi4_237-2A-GFP[17] and the *Arabidopsis RIBULOSE-1,5-BISPHOSPHATE CARBOXYLASE/OXYGENASE SMALL SUBUNIT 2b* (rbc) terminator was cloned into pDONR P4r-P3r to yield pE(R4-R3)Ppubi4-Cas10d-Cas5d-Cas7d. Multisite gateway LR reaction was done using pE(L1-L4)P1.2-Cas3d-Cas6d-GFP, pE(R4-R3) Ppubi4-Cas10d-Cas5d-Cas7d, pE(L3-L2)AtU6gRNA, and the destination binary vector pTGW12 (Supplementary Fig. 21) to yield pMGTiD20. For insertion of gRNA sequence, two oligonucleotides containing a target sequence were annealed and cloned into the gRNA expression vector using Golden Gate cloning using restriction enzyme BsaI (NEB) into the spacer sequence of pTiDP1.2 or pMGTiD20. The pTiDP1.2 construct has a single CaMV35S promoter driving all 5 ORFs of Cas effectors separated by 2 A self-cleaving peptide (Supplementary Fig. 17), and the pMGTiD20 vector has the two expression cassettes under the two promoters: CaMV35S and *Parsley ubiquitin 4-2* (Supplementary Fig. 22).

**Plasmid interference assay.** *Escherichia coli* strain BL21-AI [F- *ompT hsd*SB (rB⁻mB⁻) *gal dcm ara*J:: T7RNAP- *tet*A] (Thermo Fisher Scientific) was used in the plasmid interference assay. *E. coli* cells harboring pCmMa567d10 were grown in LB medium supplemented with chloramphenicol (30 mg/ml). To calculate the library size of 4-nt PAM, pPAMlib-ccdB was introduced into *E.coli* cells harboring

pCmMaTiD567d10, and the *E.coli* cells were plated onto LB agar medium supplemented with chloramphenicol (30 mg/mL) and kanamycin (25 mg/mL) and grown at 37 °C overnight. The amount of pPAMlib-ccdB DNA used in the PAM screening experiment was decided as that leading to the formation of approximately 26,000 colonies, which is 100 times the theoretical-library size of the 4-nt PAM. The appropriate amount of pPAMlib-ccdB DNA was introduced into *E. coli* cells harboring pCmMaTiD567d10. After transformation, the *E.coli* cells were precultured in SOB liquid medium supplemented with 0.2% arabinose at 37 °C for 2 h, then plated onto LB agar medium supplemented with 0.2% arabinose, 0.4 mM IPTG, chloramphenicol (30 mg/mL) and kanamycin (25 mg/mL) for growth at 37 °C overnight. Plasmid DNA was extracted from surviving *E. coli* colonies, and the PAM sequence was amplified with adapters for Illumina sequencing using extracted plasmid DNAs as templates. The 4-nt PAM regions from 300–400 reads were analyzed with Miseq and counted manually.

**Luc reporter assay.** Human embryonic kidney cell line 293 T (HEK293T, RIKEN BRC) was used in luc reporter assay. Cells are cultured in Dulbecco's modified Eagle's Medium (DMEM) supplemented with 10% fetal bovine serum (Thermo Fisher Scientific), GlutalMAX™ Supplement (Thermo Fisher Scientific), 100 units/mL penicillin, and 100 μg/mL streptomycin in a 60 mm dish at 37 °C with 5% $CO_2$ incubation. Cells ($2.0 \times 10^4$ cells/well) were seeded onto 96-well plates (Corning) the day before transfection and transfected using TurboFect Transfection Reagent (Thermo Fisher Scientific) following the manufacturer's protocol. A total of 200 ng plasmid DNAs including (1) pGL4.53 vector encoding *Fluc* gene (Promega) used as an internal control, (2) pCAG-nLUxxUC vector interrupted with target DNA fragment (Supplementary Table 2), (3) plasmid DNAs encoding TiD components (Supplementary Fig. 6, pEFs vectors), and (4) pAEX-hU6gRNA for the gRNA expression vector were used in each well of a 96-well plate. NanoLuc and Fluc luciferase activities were measured 3 days after transfection using the Nano-Glo® Dual-Luciferase® Reporter Assay System (Promega). The NanoLuc/Fluc ratio was calculated for each sample. The NanoLuc/Fluc ratio of the sample transfected with non-targeting gRNAs was used as the control and the relative activity was calculated for each sample to evaluate the gRNA activity. The experiments were repeated three to four times independently with similar results.

**Plant transformation.** Tomato plants (*Solanum lycopersicum* L.) cv. Micro-Tom and Ailsa Craig were used for site-directed mutagenesis experiments. Plants were grown under conditions of 24 °C with 16 h light at 4000–6000 lx/8 h dark in the growth chamber. Transgenic tomato plants were generated using TiD vectors for plants. Leaf disks from tomato cotyledons were transformed with *Agrobacterium tumefaciens* strain GV2260 harboring the TiD vector. Transgenic calli and shoots were selected and regenerated to plantlets on MS medium containing 100 μg/mL kanamycin according to the method of Ueta et al.[17].

**DNA deletion analysis by long-range PCR.** For mutation analysis by long-range PCR in tomato plants, genomic DNA were isolated independently from 20 T0 transgenic TiD tomato calli for *SlIAA9* GTC_gRNA1(+) and GTT + GTT_gRNA5 (−) (+), respectively, and 30 T0 calli and 12 T0 shoots for *SlRIN* GTC_4003–4238 (+) using NucleoSpin® Plant II (TaKaRa Bio). To analyze the large deletions, a region of about 5–6 kbp including the target site of each gRNA was amplified by first PCR for *SlIAA9* GTC_gRNA1(+) and by nested-PCR for *SlIAA9* GTT + GTT_gRNA5(−) (+) and *SlRIN* GTC_4003–4238(+) using PrimeSTAR GXL DNA Polymerase (TaKaRa Bio) and several kinds of primer sets for long-range PCR under the following conditions: 35 cycles of 10 s at 98 °C, 15 s at 60 °C, and 7 min at 68 °C. The PCR products were analyzed by 1% agarose gel electrophoresis and stained with ethidium bromide. The first round PCR products for *SlIAA9* GTC_gRNA1(+) were pooled and purified for cloning. The purified PCR products were cloned into pMD20-T vector using Mighty TA-cloning Kit (TaKaRa). Nested PCR for *SlIAA9* GTT + GTT_gRNA5(−) (+) and the *SlRIN* GTC_4003–4238(+) transgenic calli was carried out, and only small DNA fragments separated in the agarose gel were extracted and purified from the gel for further analyses. For the *SlRIN* GTC_4003–4238(+) transgenic shoots, nested PCR was performed twice using the same primer sets and the small DNA fragments were also extracted after gel electrophoresis. The cloning of extracted fragments was carried out as mentioned above. Each cloned plasmid was analyzed by Sanger sequencing. The clone numbers for sequencing were varied for each sample as described in the results. All primers used for long-range PCRs used in the mutation analyses are listed in Supplementary Table 4.

**Mutation analyses in short-range PCR products.** To evaluate mutations introduced in transfected transgenic tomato calli and shoots, a region of about 400 bp surrounding the target locus of gRNA was amplified by short-range PCR using a PCR kit as described above. In the Cel-1 assay, PCR products from transgenic plants were digested using a Surveyor® Mutation Detection Kit (IDT). In PCR-RFLP, the PCR products from transgenic tomato plants were digested with AccI. Mutated and the wild-type DNA fragments were separated by 2–2.5% agarose gel electrophoresis and stained by GelRed (Biotium). PCR amplicons were also cloned into the TA cloning vector (TaKaRa Bio) to determine their sequences by the Sanger method. Amplicon deep sequences for on- and off-targets mutation

analyses were performed using Multiplex identifiers-labeled PCR[17]. PCR products were subjected to Truseq on the MiSeq platform (Illumina). MiSeq data was analyzed using CLC Genomics Workbench software version 7.5.1 (CLC bio). All primers used for short-range PCRs used in the mutation analyses are listed in Supplementary Table 3.

**In silico analysis for TiD target sites.** Target sites of TiD and SpCas9 and the DNA sequences (on-target sequences) were mined in the tomato chromosome 4 and 5, and the whole genome of Arabidopsis and rice by an in-house Perl script, respectively. For each on-target sequence, the off-target sites with up to five mismatches were identified by using a tool Cas-OFFinder[28]. Then, the total numbers of on-target sites and off-target sites in the chromosome 4 were calculated. Similarly, on-target sites and sequences within the entire genomic regions of the *SlIAA9* and *RIN* genes were detected by an in-house Perl script, respectively. For each on-target sequence of the gene, all off-target sites were identified in the tomato genome sequence by Cas-OFFinder. The total numbers of on-target sites of the gene and off-target sites in the genome were obtained.

**Statistics and reproducibility.** Statistical significance and p values were assessed by analysis of Student's *t* tests, and error bars reflect SEM with $n \geq$ three biological triplicates. The number of replicates and samples sizes are stated in the respective Figure Legend for each figure and Methods.

**Reporting summary.** Further information on research design is available in the Nature Research Reporting Summary linked to this article.

## Data availability
All relevant data and clones from this work are available from the corresponding author upon reasonable request. Source data are in Supplementary Data 1 and 2.

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

## Acknowledgements

We thank E. Nakashima for their technical assistance. This research was supported by the New Energy and Industrial Technology Development Organization (NEDO; to K.O., K.Y., Y. O.). Tomato seed (TOMJPF00004) was provided by the National Bio-Resource Project (NBRP), AMED, Japan.

## Author contributions

K.O. and Y.O. designed and wrote the manuscript with help from all authors. N.W. designed and performed experiments using animal cells. T.M. performed experiments using plant cells to detect the long-range PCR. E.M. performed experiments using bacterial and animal cells. K.M. constructed the vectors for plant experiments. R.U., R.H. and C.A.-H. performed experiments using plant cells. B.K. and K.Y. performed in-silico analysis.

## Competing interests

The authors declare no competing interests.
