## [Peer Review File · Communications Biology]

Reviewers' comments:

Reviewer #1 (Remarks to the Author):

Osakabe et al report on the adaptation of a the CRISPR type I-D (TiD) system for editing in plants. This are exciting novel data that well deserve a wide audience as guaranteed by communication biology. The authors show that the functional nuclease Cas10d is inducing different kinds of mutations in comparison to Cas9 in the tomato genome, very interestingly also larger deletions. Biallelic mutations could be obtained. Obviously TiD will not replace Cas9, but is a valuable add to our toolbox to manipulate plant genomes.

The study is sound and carefully done. If this the first report on TiD plant gene editing it should be published without further ado. I am only concerned about reference 16 by the same authors as in the title TiD application not only in mammals but also in plants is mentioned. Can the author comment on this fact and clearly state whether or not there is an overlap with the current publication? If there is a duplication it would be indeed a major issue.

Reviewer #2 (Remarks to the Author):

In this manuscript, the author identified a Class 1 type I subtype CRISPR-Cas genomic locus, named type I-D (TiD), which consists of eight Cas genes (Cas1d–Cas7d, Cas10d) followed by an array of repeat-spacer. The small indels, long-range deletions and bi-allelic mutations were all effectively induced by the CRISPR TiD in tomato calli and regenerated shoots. From my own perspective, this research is of great importance for exploring and expanding CRISPR/Cas genome editing series tools in plant research, however, some concerns need to be addressed before publication on Communications Biology.

Major concerns:

1. This manuscript aims to explore whether CRISPR TiD is suitable for genome editing in plants. However, the TiD system was mainly based on research in mammalian cells (Reference 16), preprinted in bioRxiv, which has not been certified by peer review. Therefore, it is necessary to introduce this system detailly in this article, such as the identification of PAM preference, sgRNA length, and mechanism of action.
2. Line 76-87: this part seems to be similar with content written in Reference 16. Although this experiment was conducted by the same team, it should not repeat it again in this article. Or it's better to test the editing activity of the TiD and Cas10d nuclease function with reporter system in plant cells rather than in mammalian cells HEK293T.
3. This manuscript mainly tested GTC/GTT PAM on-target edits, whereas in off-target verification part it tested GTH (H for A, C, T) PAM. No GTA PAM was tested in on-target edits. Some necessary explanation or extra experiment on GTA PAM on-target testing should be provided.
4. Section "Detection of the long-range deletion mutations by TiD in plants" and section "Long-range deletion mutations at the RIN locus mediated by TiD in tomato" detected the long-range mutations in gene SI1AA9 and SIRIN, respectively. However, they both focus on one topic: CRISPR TiD is capable of triggering long-range deletion in tomato calli and regenerated shoots. Therefore, in my opinion, these two sections could be integrated into one section.
5. Section "Off-target effects generated by TiD in the plant genome", in sentence "We then evaluated the off-target mutations in the T0 generation of tomato plants exhibiting clear SI1AA9-gene knock-out phenotypes, selecting three potential off-target sites for SI1AA9GTC_gRNA1(+) with 9–11 mismatches that are the lowest mismatches for the target", it is relatively too long to select 9-11 bp mismatches, which affect a lot on editing efficiency. It is recommended to select the target with less mismatch (less than 5 mismatches) for re-examination to detect its off-target efficiency. Moreover, for detecting the off-target effects of the long-range deletion mutation, it is also necessary to detect whether there exist small indels at off-target sites.
6. Line 159, "These data suggest that the TiD has the advantages for genome editing." The

conclusion may be affected by various genes, chromosomes, and species. (why only chromosome 4 was selected for off-target detection.) Therefore, it cannot effectively prove this conclusion. More evidences should be provided.

Minor comments:

1. "Cas12/Cpf1" should be "Cas12a/Cpf1", please check them all in this manuscript.
2. Line 83: "35b" should be "35 bp".
3. Line 92: what's the meaning of "A (-)" in "GTT-gRNA5-A(-)" ?
4. Insertions and deletions should all be written in either "in/del" or "indels" to make it more unified.
5. Line130: Cas12a belongs to type V.
6. In reference 16, title "Genome editing in mammals and plants using CRISPR type I-D nuclease" is inconsistent with the article preprinted in bioRxiv, which was titled "Genome editing in mammals using CRISPR type I-D nuclease".

Reviewer #3 (Remarks to the Author):

This paper overall did a good job of showing that (1) Class I Type I-D systems work effectively in plants, (2) bi-directional long range deletions can be generated using type I-D systems and (3) Class I Type I-D systems may have lower off target mutation rates than class II type II systems like Cas9. This information is noteworthy and will be of interest to many. I do however have some criticisms of the paper:

Fig 2A is a bit confusing. It is stated that 7/11 calli and 14/15 shoots showed evidence of somatic editing. However, the way that data is presented is a bit messy and hard to interpret. They are mixing Cel-1 and PCR-RFLP to show evidence of editing, and seemingly not showing gel images for all 7 and 14 editing-positive calli & shoots. From reading it seems they ran the Cel-1 assay first on their calli, followed by the PCR-RFLP analysis on the regenerated shoots. While this is a good proxy for a yes/no editing answer, I have a tough time interpreting all the gel images stuck together in the same figure panel. Perhaps simplifying panel 2A by adding some/all of the Cel-1 or PCR-RFLP images to the supplemental data would be more appropriate.

As a general comment about the cumulative work shown in Figures 1+2, it would have been nice to see some quantitative transient editing data using TiD. Further, it would be nice to see how TiD compares to Cas9 and/or Cpf1

-By generating transgenic plants using selection I think this is a scenario in which you're likely going to recover several edited plants by virtue of selecting for expression of the targeted nuclease over time. In this regard, I didn't ever see any data on the number of calli that were transformed and subsequently discarded, which I am assuming likely occurred.

-The absence of any negative data here is a bit disappointing for me as well, as it paints a picture that this TiD system is close to 100% effective. It very well could be highly efficient, but not having a comparison with the current standard in the field of Cas9 makes me question the true efficiency of TiD because I don't really have an efficiency benchmark for it.

They very quickly glaze over the subtle point that their construct design changes when they mutated the RIN locus. Originally, they had a single 35S promoter driving all 5 ORFs separated by P2A tags when mutating the IAA9 gene (Ext. Data Fig 1). As we know, this system likely results in slowly decreasing levels of each individual ORF moving from the N to C terminus. They seemingly address this by using a two-promoter system wherein they drive Cas6 and Cas3 under the 35S promoter separated by P2A tags and then driving Cas10, Cas5, and Cas7 under the pUbi4 promoter separated by P2A tags. Why was this decision made? Did mutation efficiencies change when switching this expression scheme?

They state, "these results indicated that the deletion mutations generated by TiD in tomato genome were bi-directional, which together with the generation of small indel mutations by TiD, is the unique feature of TiD that differs from mutation by type I-E and type II effectors such as Cas9 and Cpf1". However, I am pretty confident that this NBT paper back in 2018 identified bi-directional larger deletions generated by Cas9 - <https://www.ncbi.nlm.nih.gov/pmc/articles/PMC6390938/>.

Another kind of semantic comment is their slight misuse of the classification of CRISPR systems. They frequently refer to their system as a 'type I' system and that Cas9/Cpf1 are 'type II' systems. This is not entirely correct. The first major phylogenetic split is between class I and class II CRISPR systems – class I systems are multi-protein complexes that carry out target recognition and cutting (such as the system described here), while class II systems utilize a single protein for target recognition and cutting. The next major splits are with various 'types' under class I and class II phylogenies. Cas9 is a class II type II nuclease, while Cpf1 is a class II type V nuclease, characterized by their slightly different nuclease domains and overall protein architecture. For class I systems, they are broken down by the different proteins which assemble in the multi-protein complex. This group frequently compares their class I type I-D system to class I type I-E. Figures 1 and 2 in this paper do provide some mostly up to date phylogenies - <https://www.nature.com/articles/s41579-019-0299-x>.

Their method for identifying off target mutations is biased. By designing PCR primers and performing an amplification of 'likely' off target regions, they are assuming their prediction of these sites is predictive of the entire genome. While this is a good proxy for off target mutations, the state of the art unbiased methods for true quantification of off target effects would be something like circle-seq from Keith Joung's group. I know this does take significantly more time and money, but it is a significant limitation of this group's analysis to extrapolate a biased analysis of selected regions vs an unbiased genome-wide analysis.

Response letter: COMMSBIO-20-0799A

#####

Responses to Reviewer 1's Comments

<Comment>

Osakabe et al report on the adaption of a the CRSPR type I-D (TiD) system for editing in plants. This are exciting novel data that well deserve a wide audience as guaranteed by communication biology. The authors show that the functional nuclease Cas10d is inducing different kinds of mutations in comparison to Cas9 in the tomato genome, very interestingly also larger deletions. Biallelic mutations could be obtained. Obviously TiD will not replace Cas9, but is a valuable add to our toolbox to manipulate plant genomes.

<Response>

Thank you very much for your positive evaluation of our manuscript. We also very much appreciate your constructive comments, which have helped us improve the quality of the manuscript.

<Comment1>

The study is sound and carefully done. If this the first report on TiD plant gene editing it should be published without further ado. I am only concerned about reference 16 by the same authors as in the title TiD application not only in mammals but also in plants is mentioned. Can the author comment on this fact and clearly state whether or not there is an overlap with the current publication? If there is a duplication it would be indeed a major issue.

<Response 1>

Thank you very much for your comments. First, we apologize that we mistakenly wrote the title of reference #16 in the previous manuscript as “Genome editing in mammals and plants using CRISPR type I-D nuclease” the correct title is “Genome editing in mammals using CRISPR type I-D nuclease”. In fact, we submitted reference #16 of the previous manuscript to bioRxiv showing TiD genome editing only for mammalian cells and not for plants. bioRxiv is a preprint paper, and #16 of the previous manuscript has not been certified by peer review.

We also apologize for one additional error in the description in the *Results and Discussion* pertaining to the results shown in Fig. **1b** in the previous version of our manuscript (now Fig. **1d** of the revised manuscript). This result indicated that, in the TiD complex, both Cas3d and Cas10d were essential for genome editing activity; however, the text in the *Results and Discussion* did not make this clear. Instead, there was some other descriptions about dCas10d (Lines 82–84 in the previous version of our manuscript). We can see how this has led to confusion. The corrected explanations for **Fig 1d** (former Fig. 1b) have been described in L.102–104 in the revised manuscript. Certainly, there are no overlaps between the bioRxiv paper for mammals and the original version of this manuscript for plants, except for this mistake.

In addition, we revised the structure of Fig. 1 to show the basic nature of the TiD system in detail in this article, because this manuscript is the first to feature TiD as a peer-reviewed paper; the identification of PAM preference and sgRNA length are now shown in Fig1b and 1e, respectively, in the revised manuscript.

We deleted citation of the bioRxiv paper (ref #16 in the previous manuscript) from the revised manuscript, because part of the results in the bioRxiv preprint are now described in this manuscript as mentioned above. Thus, the results in the bioRxiv paper are now divided into two papers: this revised manuscript and another new one that will describe TiD activity only for mammalian cells (currently in

preparation, not yet submitted or published in any journal). This latter new manuscript does not overlap with any experiments and results in the current revised manuscript submitted here, which focuses mainly on plants.

Together, the Introduction, the first section “**TiD composed of Cas effector proteins with a Cas10d can be used for genome editing**” of descriptions about Fig. 1, and Methods were revised as follows;

L.63-66;

“In addition, one of major Cascade factors for PAM recognition, the Cas8 homologous protein¹⁶ is also missing in TiD. Therefore, the mode of interference through PAM recognition to target DNA degradation in the TiD system will differ from that of other type I systems.”

L.79-94 (Results and Discussion) ;

“The typical Cas8 gene—the common effector in CRISPR type I-A, B, C, E, and F^{2,16}—is missing from the CRISPR/Cas TiD locus of *M. aeruginosa*, predicting different mechanisms of cascade complex stability and *in vivo* DNA cleavage activity in TiD compared with other type I sub-types (**Fig. 1a**). To identify the PAM in the TiD system in *M. aeruginosa*, we performed a depletion assay using the negative selection marker *ccdB*. pCmMa567d10 (**Supplemental Fig. 6**), containing expression cassettes for Cas5d, Cas6d, Cas7d and Cas10d carrying a mutation in the HD-like domain [dCas10d (H177A)] and gRNAs targeted to the *ccdB* promoter, was introduced into *Escherichia coli* strain BL21-AI followed by a PAM library plasmid, pPAMlib-*ccdB* (**Fig. 1b, Supplemental Fig. 6**). *ccdB* negative selection revealed the PAM: 5'-GTH-3' (H=A or C or T) adjacent to the target sequence (**Fig. 1b, lower panel**). When pCmMa567 (**Supplemental Fig. 6**), which carries expression cassettes for Cas5d, Cas6d and Cas7d, was used for screening instead of pCmMa567d10, GTH PAMs were screened out, but the resulting transformants were unstable, and growth of the *E. coli* cells was very weak. These results suggested that Cas10d requires the correct PAM for full repression, and that Cas10d is a functional counterpart of Cas8 for PAM recognition and stabilization. We did not find any similar amino acid sequences shared between the Cas10d and Cas8 protein families.”

L.98-110 (Results and Discussion) ;

“First, the human *AAVSI* gene fragment containing the TiD target site (**Supplementary Table S2**) was used to evaluate the TiD complex using the 35-bp crRNA spacer sequence. In this assay, HEK293T cells were transfected simultaneously with TiD Cas effectors, gRNA, and NanoLuc interrupted with target gene fragment and firefly luciferase expression vectors, and then the luc reporter assay was carried out 72 hr after transfection. Deletion of either Cas3d or Cas10d abolished TiD genome editing activity in the luc reporter assay (**Fig. 1d**), suggesting that both Cas3d or Cas10d have essential roles in genome editing activity. In the original CRISPR locus of *M. aeruginosa* strain PCC9808, both 35-bp and 36-bp spacer sequences are used to target specific genomic DNAs; both spacers function in genome editing in human cells (**Fig. 1e**). To evaluate the genome editing activity of TiD for plant genes, we next performed the luc reporter assay for several target rice and tomato gene sequences, *SIIAA9* (important in parthenocarpy)¹⁷ and *NADK2* (*OsNADK2*) (**Fig. 1f, Supplementary Fig. 1b**). The results showed that there were several targets with GTT or GTC PAM with higher activity in the luc reporter assay than GTA PAM.”

L.270-285 (Methods) ;

“**Bacterial vectors:** Gene fragments corresponding to.... and inserted in front of the *lacZ-ccdB* gene of pPAM-*ccdB*.”

L.323-339 (Methods) ;

“**Plasmid interference assay** *Escherichia coli* strain BL21-AI... with Miseq and counted manually.”

#####

Responses to Reviewer 2's Comments

<Comment>

In this manuscript, the author identified a Class 1 type I subtype CRISPR-Cas genomic locus, named type I-D (TiD), which consists of eight Cas genes (Cas1d–Cas7d, Cas10d) followed by an array of repeat-spacer. The small indels, long-range deletions and bi-allelic mutations were all effectively induced by the CRISPR TiD in tomato calli and regenerated shoots. From my own perspective, this research is of great importance for exploring and expanding CRISPR/Cas genome editing series tools in plant research, however, some concerns need to be addressed before publication on Communications Biology.

<Response>

Thank you very much for your positive evaluation of our manuscript. We also very much appreciate your constructive comments, which have helped us improve the quality of the manuscript. We addressed your major comments as follows:

<Major comment 1>

This manuscript aims to explore whether CRISPR TiD is suitable for genome editing in plants. However, the TiD system was mainly based on research in mammalian cells (Reference 16), preprinted in bioRxiv, which has not been certified by peer review. Therefore, it is necessary to introduce this system detail in this article, such as the identification of PAM preference, sgRNA length, and mechanism of action.

<Response to comment 1>

Thank you very much for your valuable comments and suggestions. According to your comment and suggestion, we rearranged the article and revised the structure of **Fig. 1** to show the basic properties of the TiD system in more detail; identification of PAM preference, sgRNA length, and discussion of the mechanisms are now shown in **Fig1b, 1e**, and in the main text.

We apologize for the mistake in the text describing the result in **Fig. 1b** in the previous manuscript (**Fig. 1d** in the revised manuscript). This result indicated that, in the TiD complex, both Cas3d and Cas10d were essential for genome editing activity; however, the text in *Results and Discussion* did not explain this result, instead including other descriptions about dCas10d (Line 82–84 in the previous manuscript). The corrected explanations for **Fig 1d** (former **Fig. 1b**) are now described in **L.102–104** in the revised manuscript.

We deleted citation of the bioRxiv paper (ref #16 in the previous manuscript) from the revised manuscript, because part of the results in the bioRxiv preprint are now described in this manuscript as mentioned above. Thus, the results in the bioRxiv paper are now divided into two papers: this revised manuscript and another new one that will describe TiD activity only for mammalian cells (currently in preparation, not yet submitted or published in any journal). This latter new manuscript does not overlap with any experiments and results in the current revised manuscript submitted here, which focuses mainly on plants.

Together, the introduction part, the first section “**TiD composed of Cas effector proteins with a Cas10d can be used for genome editing**” of descriptions about **Fig. 1**, and Methods were revised as follows:

L.63-66;

“In addition, one of major Cascade factors for PAM recognition, Cas8 homologous protein¹⁶ is also missing in TiD. Therefore, the mode of interference through PAM recognition to target DNA

degradation in the TiD system will differ from that of other type I systems.”

L.79-94 (Results and Discussion) ;

“The typical Cas8 gene—the common effector in CRISPR type I-A, B, C, E, and F^{2,16}—is missing from the CRISPR/Cas TiD locus of *M. aeruginosa*, predicting different mechanisms of cascade complex stability and *in vivo* DNA cleavage activity in TiD compared with other type I sub-types (**Fig. 1a**). To identify the PAM in the TiD system in *M. aeruginosa*, we performed a depletion assay using the negative selection marker *ccdB*. pCmMa567d10 (**Supplemental Fig. 6**), containing expression cassettes for Cas5d, Cas6d, Cas7d and Cas10d carrying a mutation in the HD-like domain [dCas10d (H177A)] and gRNAs targeted to the *ccdB* promoter, was introduced into *Escherichia coli* strain BL21-AI followed by a PAM library plasmid, pPAMlib-*ccdB* (**Fig. 1b, Supplemental Fig. 6**). *ccdB* negative selection revealed the PAM: 5'-GTH-3' (H=A or C or T) adjacent to the target sequence (**Fig. 1b, lower panel**). When pCmMa567 (**Supplemental Fig. 6**), which carries expression cassettes for Cas5d, Cas6d and Cas7d, was used for screening instead of pCmMa567d10, GTH PAMs were screened out, but the resulting transformants were unstable, and growth of the *E. coli* cells was very weak. These results suggested that Cas10d requires the correct PAM for full repression, and that Cas10d is a functional counterpart of Cas8 for PAM recognition and stabilization. We did not find any similar amino acid sequences shared between the Cas10d and Cas8 protein families.”

L.98-110 (Results and Discussion) ;

“First, the human *AAVSI* gene fragment containing the TiD target site (**Supplementary Table S2**) was used to evaluate the TiD complex using the 35-bp crRNA spacer sequence. In this assay, HEK293T cells were transfected simultaneously with TiD Cas effectors, gRNA, and NanoLuc interrupted with target gene fragment and firefly luciferase expression vectors, and then the luc reporter assay was carried out 72 hr after transfection. Deletion of either Cas3d or Cas10d abolished TiD genome editing activity in the luc reporter assay (**Fig. 1d**), suggesting that both Cas3d or Cas10d have essential roles in genome editing activity. In the original CRISPR locus of *M. aeruginosa* strain PCC9808, both 35-bp and 36-bp spacer sequences are used to target specific genomic DNAs; both spacers function in genome editing in human cells (**Fig. 1e**). To evaluate the genome editing activity of TiD for plant genes, we next performed the luc reporter assay for several target rice and tomato gene sequences, *SlIAA9* (important in parthenocarpy)¹⁷ and *NADK2* (*OsNADK2*) (**Fig. 1f, Supplementary Fig. 1b**). The results showed that there were several targets with GTT or GTC PAM with higher activity in the luc reporter assay than GTA PAM.”

L.270-285 (Methods) ;

“**Bacterial vectors:** Gene fragments corresponding to.... and inserted in front of the *lacZ-ccdB* gene of pPAM-*ccdB*.”

L.323-339 (Methods) ;

“**Plasmid interference assay** *Escherichia coli* strain BL21-AI... with Miseq and counted manually.”

<Major comment 2>

Line 76-87: this part seems to be similar with content written in Reference 16. Although this experiment was conducted by the same team, it should not repeat it again in this article. Or it's better to test the editing activity of the TiD and Cas10d nuclease function with reporter system in plant cells rather than in mammalian cells HEK293T.

<Response to comment 2>

Thank you very much for your valuable comments and suggestions. First, we are sorry that there was a

mistake in the text for the result of Fig. **1b** in the previous manuscript as mentioned above.

We rearranged this part, adding the data to address your Major Comment 1 to show the basic properties of the TiD system in detail in this article as mentioned above. These additional results have previously been presented in the reference #16, a bioRxiv preprint, but we adopted them in this article in accordance with your comment.

Again, we would like to mention that we deleted citation of the bioRxiv paper (ref #16 in the previous manuscript) from the revised manuscript, because part of the results in the bioRxiv preprint are now described in this manuscript as mentioned above. Thus, the results in the bioRxiv paper are now divided into two papers: this revised manuscript and another new one that will describe TiD activity only for mammalian cells (currently in preparation, not yet submitted or published in any journal). This latter new manuscript does not overlap with any experiments and results in the current revised manuscript submitted here, which focuses mainly on plants.

We also thank you for your suggestion regarding the reporter system in plant cells, and, according with your suggestion, we are developing the system to evaluate the sgRNA in plant cells and hope to show data describing details of the TiD system, especially the evaluation of more targets for plants in the next manuscript.

<Major comment 3>

This manuscript mainly tested GTC/GTT PAM on-target edits, whereas in off-target verification part it tested GTH (H for A, C, T) PAM. No GTA PAM was tested in on-target edits. Some necessary explanation or extra experiment on GTA PAM on-target testing should be provided.

<Response to comment 3>

Thank you very much for your comment and suggestion. In the luc reporter assay, we also tested several rice targets with GTA PAM for another gene. In accordance with your comment, we have added this result in **Supplementary Fig 1b** in the revised manuscript. The results showed the relatively low activity for GTA PAM, therefore, we have examined the GTC PAM and GTT PAM for genome editing of endogenous targets in plants. We added a description of these data in the main text (**L.106–110**). We also rearranged descriptions of the TiD vector for plants (L88–94 in the previous version of manuscript) and transferred them to the section “**Targeted mutagenesis by TiD in plants**”.

L.106-110;

“To evaluate the genome editing activity of TiD for plant genes, we next performed the luc reporter assay for several target rice and tomato gene sequences, *SIIAA9* (important in parthenocarpy)¹⁷ and *NADK2* (*OsNADK2*) (**Fig. 1f, Supplementary Fig. 1b**). The results showed that there were several targets with GTT or GTC PAM with higher activity in the luc reporter assay than GTA PAM.”

<Major comment 4>

Section “Detection of the long-range deletion mutations by TiD in plants” and section “Long-range deletion mutations at the RIN locus mediated by TiD in tomato” detected the long-range mutations in gene *SIIAA9* and *SIRIN*, respectively. However, they both focus on one topic: CRISPR TiD is capable of triggering long-range deletion in tomato calli and regenerated shoots. Therefore, in my opinion, these two sections could be integrated into one section.

<Response to comment 4>

Thank you very much for your comment and suggestion. We rearranged the sections based on your comment; we integrated the section “Detection of the long-range deletion mutations by TiD in plants”

and the section “Long-range deletion mutations at the RIN locus mediated by TiD in tomato” into a single section “**Detection of long-range deletion mutations by TiD in plants**” (L.159–196) in the revised manuscript.

<Major comment 5>

Section “Off-target effects generated by TiD in the plant genome”, in sentence “We then evaluated the off-target mutations in the T0 generation of tomato plants exhibiting clear *SIIAA9*-gene knock-out phenotypes, selecting three potential off-target sites for *SIIAA9*GTC_gRNA1(+) with 9–11 mismatches that are the lowest mismatches for the target”, it is relatively too long to select 9-11 bp mismatches, which affect a lot on editing efficiency. It is recommended to select the target with less mismatch (less than 5 mismatches) for re-examination to detect its off-target efficiency. Moreover, for detecting the off-target effects of the long-range deletion mutation, it is also necessary to detect whether there exist small indels at off-target sites.

<Response to comment 5>

Thank you very much for your important comments and suggestions. We agree with your points that, to evaluate off-targets, the experiments should be carried out using candidates with less mismatches to detect its efficiency. In this manuscript, we used candidates for off-target effects that had 9–11 and 6–7 mismatches for *SIIAA9* and *SIRIN*, respectively, as the *in-silico* selected candidates that have fewest mismatches for these on-target sequences. Unfortunately, we currently do not have any other good on-targets, which show the high activity of genome editing in plant cells and have several off-target candidates with less than 5 mismatches as well. Considering these criteria, at first we need additional selection for on-targets with high activity, and then further experiments need to be carried out to show the whole results including the off-target analysis and development of mutant plants as described for the *SIIAA9* and *RIN* genes in this manuscript. Furthermore, as shown in **Fig. 5a, 5b** and **Supplementary Fig. 4**, since there are lower number of off-target sequences for TiD in the genome, many candidates would need to be evaluated to select a representative one that has off-targets with fewer mismatches. In accordance with the important point you make, therefore, we would like to add a description in the main text of the revised manuscript to explain that on-targets with high activity and off-targets with fewer mismatches need to be tested for TiD off-target efficiency in plant cells (L.214–236).

Thank you very much for your important comments regarding off-targets and small indels. In accordance with your comments, we added data to evaluate whether small indels exist at off-target sites in **Supplementary Fig. 5c** and a description for the results in which there were no small indels.

Together, the latter half of the section “**Off-target effects generated by TiD in the plant genome**” was revised as follows;

L.214-236;

“Although the gRNA target sequences used in this study, *SIIAA9* GTC_gRNA1(+), *SIIAA9* GTT+GTT_gRNA5(-)(+), and *SIRIN* GTC_4003–4238(+), do not have highly similar sequences, and have fewer mismatches in the tomato genome, we next evaluated the off-target mutations in the T0 generation of tomato plants exhibiting clear *SIIAA9*-gene knock-out phenotypes; Three potential off-target sites for *SIIAA9* GTC_gRNA1(+) with 9–11 mismatches, two potential off-target sites for *SIIAA9* GTT+GTT_gRNA5(-)(+) with 11 mismatches, and two potential off-target sites for *SIRIN* GTC_4003–4238(+) with 6 and 7 mismatches, respectively, which are the sites with lowest mismatches for each on-target, were selected and further analyzed (**Fig. 5c, 5d, Supplementary Table. 5, Supplementary Fig. 5**). MiSeq analysis of PCR products around the potential off-target sites for

SIIAA9 GTC_gRNA1(+) showed that there was little-to-no off-target mutation in the T0 generation of tomato plants (Fig. 5c). Long-range nested PCR of the potential off-targets for *SIIAA9* GTC_gRNA1(+) and *SIIAA9* GTT+GTT_gRNA5(-)(+) was also performed using specific primers located around 5 – 8 kbp upstream and downstream of the target sequence, and the results suggested there were no obvious effects (Supplementary Fig. 5a, b). Also, the off-target effects of long-range deletion mutations were evaluated for *SIRIN* GTC_4003–4238(+) in the T0 transgenic plants using specific primers located around 3 kbp upstream and downstream of the target sequence, respectively. As before, no off-target mutations were found in the T0 generation of tomato plants (Fig. 5d). The Cel-I assay to evaluate small indels also showed no digested bands in the *SIIAA9* GTT+GTT_gRNA5(-)(+) and *SIRIN* GTC_4003–4238(+) lines, indicating no mutations in these off-target sites (Supplementary Fig. 5c). Together with a comprehensive analysis of many other on-targets for TiD, further work *in vivo* to evaluate off-target effects for fewer mismatches will be required in order to precisely elucidate the mechanisms of the TiD system when used in conjunction with the advanced unbiased technologies, i.e. CIRCLE-seq²⁰ and DISCOVER-seq²¹.”

<Major comment 6>

Line 159, “These data suggest that the TiD has the advantages for genome editing.” The conclusion may be affected by various genes, chromosomes, and species. (why only chromosome 4 was selected for off-target detection.) Therefore, it cannot effectively prove this conclusion. More evidences should be provided.

<Response to comment 6>

Thank you for your comment and suggestion. We analyzed TiD target and off-target sequences in tomato chr4 as one example from a plant genome in the original manuscript. To address your points, we added data from the whole genome of Arabidopsis and rice, and tomato chr5 as another example for the tomato genome. These data indicate similar conclusions in different plant species; the potential off-targets for TiD are less than those for Cas9, although the on-target sequences for TiD in rice are less than those for Cas9 due to high GC content. The data have been added in Fig. 5b and Supplementary Fig. 4 of the revised manuscript. We also revised the main text as follows:

L. 198-213;

“Off-target effects generated by TiD in the plant genome

We next analyzed TiD off-target effects in plant genome. The TiD targets that has the 5'- GTH -3' PAM in the whole genome of Arabidopsis and rice, and entire region of tomato chromosome 4 and 5, and each *SIIAA9* and *SIRIN* gene were counted and compared to those of Cas9 (5'- NGG -3' PAM) (Fig. 5a, b, Supplementary Fig. 4, on-target). In this analysis, tomato chromosomes were selected as being representative of the tomato whole genome. The results indicate the more target sites for TiD exist in both the target genes and chromosome levels in tomato and Arabidopsis than those for Cas9. On the contrary, the rice genome has more Cas9 targets than those of TiD, this might result from the higher GC content in the rice genome and Cas9 PAM than in other species and the TiD PAM. Furthermore, when the off-target candidate sequences which contain 0 to 5 mismatches were also counted in tomato whole genome for each *SIIAA9* and *SIRIN* gene, in Arabidopsis and rice whole genomes, and in the representative chromosomes of tomato for the on-targets in the same chromosome, respectively, there are less mismatch sequences for TiD than those for Cas9 (Fig. 5a, b, Supplementary Fig. 4, mismatch numbers 0 - 5). In rice chromosomes, the decreasing tendency of TiD targets compared with Cas9 was clearer in off-targets. These data show that there are less off-target sequences for TiD in plant genomes, suggesting a TiD advantage in plant genome editing.”

L.400-403 (Methods) ;

“*in silico* analysis for TiD target sites

Target sites of TiD and SpCas9 and the DNA sequences (on-target sequences) were mined in the **tomato chromosome 4 and 5, and the whole genome of Arabidopsis and rice** by an in-house Perl script, respectively.”

<Minor comment1,2>

1. “Cas12/Cpf1” should be “Cas12a/Cpf1”, please check them all in this manuscript.

2. Line 83: “35b” should be “35 bp”.

<Response to Minor comment1&2>

Thank you very much these comments. We corrected them.

<Minor comment3>

3. Line 92: what’s the meaning of “A (-)” in “GTT-gRNA5-A(-)” ?

<Response to Minor comment3>

GTT_gRNA5-A(-) and GTT_gRNA5-B(+) were used in the multiplex CRISPR/Cas9 vector.

Supplementary Table S1 shows the sequences; (+) means sense strand and (-) means antisense strand (we added the explanation to the footnote of **Supplementary Table S1** in the revised manuscript).

We also added the explanation “TiD CRISPR crRNA (5'-GTTCCAATTAATCTTAAGCCCTATTAGGGATTGAAAC-3') was inserted between two gRNAs, GTT_gRNA5-A(-) and GTT_gRNA5-B(+), in the multiplex vector, in which two gRNAs can be expressed separately in plant cells” in **Supplementary Table S1**.

<Minor comment4>

4. Insertions and deletions should all be written in either “in/del” or “indels” to make it more unified.

<Response to Minor comment4>

We revised “insertions and deletions” to “indels”.

<Minor comment5>

5. Line130: Cas12a belongs to type V.

<Response to Minor comment5>

Thank you very much for this comment. We revised this part as pointed out to describe the recent Cas9 study, in which Cas9 induced rare complex large deletions (**L173–175**). We carefully described the classes and types of CRISPR in the introduction part in the revised manuscript (**L41–43, L49**).

<Minor comment6>

In reference 16, title “Genome editing in mammals and plants using CRISPR type I-D nuclease” is inconsistent with the article preprinted in bioRxiv, which was titled “Genome editing in mammals using CRISPR type I-D nuclease”.

<Response to Minor comment6>

Thank you very much for this comment. We are sorry for this mistake.

We now deleted this reference as mentioned above, in Response to Major Comments 1 and 2.

#####

Responses to Reviewer 3's Comments

<Comment>

This paper overall did a good job of showing that (1) Class I Type I-D systems work effectively in plants, (2) bi-directional long range deletions can be generated using type I-D systems and (3) Class I Type I-D systems may have lower off target mutation rates than class II type II systems like Cas9. This information is noteworthy and will be of interest to many.

I do however have some criticisms of the paper:

<Response>

Thank you very much for your positive evaluation of our manuscript. We very much appreciate your constructive comments, which have helped us improve the quality of the manuscript. We addressed your valuable comments as follows;

<Major comment 1>

Fig 2A is a bit confusing. It is stated that 7/11 calli and 14/15 shoots showed evidence of somatic editing. However, the way that data is presented is a bit messy and hard to interpret. They are mixing Cel-1 and PCR-RFLP to show evidence of editing, and seemingly not showing gel images for all 7 and 14 editing-positive calli & shoots. From reading it seems they ran the Cel-1 assay first on their calli, followed by the PCR-RFLP analysis on the regenerated shoots. While this is a good proxy for a yes/no editing answer, I have a tough time interpreting all the gel images stuck together in the same figure panel. Perhaps simplifying panel 2A by adding some/all of the Cel-1 or PCR-RFLP images to the supplemental data would be more appropriate.

<Response to comment 1>

Thank you very much for your comments and suggestions; we are sorry that the data in **Fig. 2a** were confusing. In accordance with your comments, we have rearranged **Fig. 2** and the description in the main text (section “**Targeted mutagenesis by TiD in plants**”, **L.112 - 144**) and added all the data for Cel-1 or PCR-RFLP to **Supplemental Fig. S2** in the revised manuscript. In fact, we carried out the Cel-1 assay first on calli, followed by PCR-RFLP analysis on regenerated shoots

<Major comment 2>

As a general comment about the cumulative work shown in Figures 1+2, it would have been nice to see some quantitative transient editing data using TiD. Further, it would be nice to see how TiD compares to Cas9 and/or Cpf1

-By generating transgenic plants using selection I think this is a scenario in which you're likely going to recover several edited plants by virtue of selecting for expression of the targeted nuclease over time. In this regard, I didn't ever see any data on the number of calli that were transformed and subsequently discarded, which I am assuming likely occurred.

-The absence of any negative data here is a bit disappointing for me as well, as it paints a picture that this TiD system is close to 100% effective. It very well could be highly efficient, but not having a comparison with the current standard in the field of Cas9 makes me question the true efficiency of TiD because I don't really have an efficiency benchmark for it.

<Response to comment 2>

Thank you very much for your comment and suggestion. We agree with your comment that it is

important to describe TiD activity in more detail, including negative data. We added the PAM frequencies in bacteria and also discussed the luc reporter assay for genome editing in plant cells with the additional target in the revised manuscript (**Fig. 1f, Supplementary Fig. 1b**). The luc reporter assay shows the quantitative transient editing data, which indicated the low activity of TiD in GTA PAM. This might be negative data, however, we think it further experiment using many targets is needed. The additional descriptions about the luc reporter assay were as follows;

L.106-110;

“To evaluate the genome editing activity of TiD for plant genes, we next performed the luc reporter assay for several target rice and tomato gene sequences, *SIIAA9* (important in parthenocarp)¹⁷ and *NADK2* (*OsNADK2*) (**Fig. 1f, Supplementary Fig. 1b**). The results showed that there were several targets with GTT or GTC PAM with higher activity in the luc reporter assay than GTA PAM.”

To address your Major Comment1, we also added all data in calli in the revised manuscript (**Supplementary Fig. 2**). Furthermore, we compared data for TiD and Cas9 that have previously been reported by us (Ueta et al) and added descriptions as follows:

L.145-157;

“Together with the *SIIAA9* experiments, TiD induced small indels at the target site (**Fig. 2**). In the previous study by Ueta et al., the CRISPR/Cas9 that targeted the *SIIAA9* exon2 – located very near the site of the *SIIAA9* GTC_gRNA1(+) – induced biallelic mutations in tomato calli¹⁷. When comparing the mutation frequencies of CRISPR/Cas9 and TiD in calli, the TiD activities were slightly lower than those of Cas9 (63.6% for TiD and 73.0% for Cas9)¹⁷; however, the TiD *SIIAA9* GTC_gRNA1(+) could not induce biallelic mutations in calli (**Fig.1a, Supplementary Fig. 2a**). Thus, TiD activity in inducing somatic mutations in calli was lower than that of Cas9. On the contrary, in the shoot samples, TiD could induce biallelic mutations at target sites with efficiency levels similar to those of Cas9 (**Fig. 2b, 2c, Supplementary Fig. 2, Supplementary Fig. 3**)¹⁷. Together, these results suggest there might be tissue specificity in the mechanism of TiD-mediated mutagenesis. Further analyses of other targets will be required to test this hypothesis; for example, detecting mutation patterns in cell lineages during shoot regeneration, and investigating tissue-specific mutagenesis might provide clues to further improvement of the TiD system in plant genome editing. ”

Furthermore, we also added the following sentence to clarify the varied mutation activities in each target for TiD vectors.

L.194-196;

“From these results, we can see that the small indels were not detected in these loci using *SIIAA9* GTT+GTT_gRNA5(-)(+) and *SIRIN* GTC_4003–4238(+), indicating that varied mutation patterns were induced by each gRNA in the CRISPR TiD system.”

<Major comment 3>

They very quickly glaze over the subtle point that their construct design changes when they mutated the RIN locus. Originally, they had a single 35S promoter driving all 5 ORFs separated by P2A tags when mutating the IAA9 gene (Ext. Data Fig 1). As we know, this system likely results in slowly decreasing levels of each individual ORF moving from the N to C terminus. They seemingly address this by using a two-promoter system wherein they drive Cas6 and Cas3 under the 35S promoter separated by P2A tags and then driving Cas10, Cas5, and Cas7 under the pUbi4 promoter separated by P2A tags. Why was this decision made? Did mutation efficiencies change when switching this expression scheme?

<Response to comment 3>

Thank you very much for your comment. As the reviewer pointed out, the reason for having two cassettes in the 2nd version of the TiD vector is that we thought the single cassettes using one promoter driving all 5 ORFs separated by P2A might affect expression levels (downregulation) of ORF(s) at the C terminus. We did not estimate this in vivo in this study. In fact, both vectors induced mutations at the targets in the tomato genome, as shown in the results of this manuscript. We think further improvement of the TiD vector would still be an important point to develop this efficient tool further.

To clarify these points, we added explanations in the revised manuscript as follows;

L.115-120;

“TiDP1.2, an all-in-one vector, harboring a single *CaMV35S* promoter driving all 5 ORFs of Cas effectors separated by 2A self-cleaving peptide, and pMGTiD20, in which two expression cassettes under two promoters (*CaMV35S* and *Parsley UBIQUITIN 4-2*) are used to express Cas effector genes (**Supplementary Fig. 1a**). The separated cassettes in pMGTiD20 were designed to eliminate decreasing C-terminal expression levels in the long single cassette for multiple ORFs in pTiDP1.2.”

L.124;

“The TiD vector, TiDP1.2, containing the designed gRNAs...”

L.179;

“..., using pMGTiD20 vector, and analyzed mutations in the transgenic calli and regenerated shoots...”

L.254-256;

“In this study, we used two types of TiD vectors, both of which induced mutations at their respective targets in the tomato genome. Further improvement of the TiD vector will still be important in developing this efficient tool.”

L.318-321;

“The TiDP1.2 construct has a single *CaMV35S* promoter driving all 5 ORFs of Cas effectors separated by 2A self-cleaving peptide, and the pMGTiD20 vector has the two expression cassettes under the two promoters: *CaMV35S* and *Parsley ubiquitin 4-2* (**Supplementary Fig. 1**).”

<Major comment 4>

They state, "these results indicated that the deletion mutations generated by TiD in tomato genome were bi-directional, which together with the generation of small indel mutations by TiD, is the unique feature of TiD that differs from mutation by type I-E and type II effectors such as Cas9 and Cpf1". However, I am pretty confident that this NBT paper back in 2018 identified bi-directional larger deletions generated by Cas9 - <https://www.ncbi.nlm.nih.gov/pmc/articles/PMC6390938/>.

<Response to comment 4>

Thank you very much for your comment and suggestion.

In accordance with your comment and suggestion, we added a description of the larger bi-directional deletion induced by Cas9, and cited this paper in the revised manuscript. We also discussed finding that the frequencies of the larger bi-directional deletion by Cas9 are lower than those by TiD, and that Cas9 induces mainly in/del mutations as follows:

L.173-175;

“... , although recent work suggested that Cas9 induced rare complex large deletions in addition to the desired small indels in mouse ES cells, and this large deletion was bi-directional as similar as TiD but with low frequency¹⁹”

<Major comment 5>

Another kind of semantic comment is their slight misuse of the classification of CRISPR systems. They frequently refer to their system as a ‘type I’ system and that Cas9/Cpf1 are ‘type II’ systems. This is not entirely correct. The first major phylogenetic split is between class I and class II crispr systems – class I systems are multi-protein complexes that carry out target recognition and cutting (such as the system described here), while class II systems utilize a single protein for target recognition and cutting. The next major splits are with various ‘types’ under class I and class II phylogenies. Cas9 is a class II type II nuclease, while Cpf1 is a class II type V nuclease, characterized by their slightly different nuclease domains and overall protein architecture. For class I systems, they are broken down by the different proteins which assemble in the multi-protein complex. This group frequently compares their class I type I-D system to class I type I-E. Figures 1 and 2 in this paper do provide some mostly up to date phylogenies - <https://www.nature.com/articles/s41579-019-0299-x>.

<Response to comment 5>

Thank you for your comment and suggestion We carefully described the classes and types of CRISPR in the introduction part in the revised manuscript (L41–43, L49).

<Major comment 6>

Their method for identifying off target mutations is biased. By designing PCR primers and performing an amplification of ‘likely’ off target regions, they are assuming their prediction of these sites is predictive of the entire genome. While this is a good proxy for off target mutations, the state of the art unbiased methods for true quantification of off target effects would be something like circle-seq from Keith Joung’s group. I know this does take significantly more time and money, but it is a significant limitation of this groups analysis to extrapolate a biased analysis of selected regions vs an unbiased genome-wide analysis.

<Response to comment 6>

Thank you for your important comments and suggestions. We agree with your comments regarding state-of-the-art unbiased methods of off-target effects, like CIRCLE-seq from Keith Joung’s group, would be powerful, allowing evaluation of the analysis with true quantification. We added discussion of these points in the revised manuscript as follows. We would like to adopt these advanced technologies to evaluate off-targets in whole genomes in the future.

L.233-2368;

“Together with comprehensive analysis with many other on-targets for TiD, further work *in vivo* to evaluate the off-target effects for less mismatches will be required for the precise elucidation of the mechanisms of TiD system also using the advanced unbiased technologies, i.e. CIRCLE-seq²⁰ and DISCOVER-seq²¹.”

REVIEWERS' COMMENTS:

Reviewer #1 (Remarks to the Author):

The authors addressed my concerns. Indeed, this revision is a quite significant improvement of the paper. I am happy to recommend publication asap.

Reviewer #2 (Remarks to the Author):

In the revised version of the manuscript, the authors had added additional data and answered the questions that I raised. I do not have further questions.

Reviewer #3 (Remarks to the Author):

The authors carefully considered my comments and suggestions. I believe the revised manuscript is now suitable for publication.